

# Comparative evaluation of approaches & tools for effective security testing of Web applications

Sana Qadir[1], Eman Waheed[1], Aisha Khanum[1] and Seema Jehan[2]

[1] Faculty of Computing, National University of Sciences & Technology, Islamabad, Pakistan
[2] Department of Computer Science, University of York, York, United Kingdom

Corresponding author
Sana Qadir, sana.qadir@seecs.edu.pk

## ABSTRACT

It is generally accepted that adopting both static application security testing (SAST) and dynamic application security testing (DAST) approaches is vital for thorough and effective security testing. However, this suggestion has not been comprehensively evaluated, especially with regard to the individual risk categories mentioned in Open Web Application Security Project (OWASP) Top 10:2021 and common weakness enumeration (CWE) Top 25:2023 lists. Also, it is rare to find any evidence-based recommendations for effective tools for detecting vulnerabilities from a specific risk category or severity level. These shortcomings increase both the time and cost of systematic security testing when its need is heightened by increasingly frequent and preventable incidents. This study aims to fill these gaps by empirically testing seventy-five real-world Web applications using four SAST and five DAST tools. Only popular, free, and open-source tools were selected and each Web application was scanned using these nine tools. From the report generated by these tools, we considered two parameters to measure effectiveness: count and severity of the vulnerability found. We also mapped the vulnerabilities to OWASP Top 10:2021 and CWE Top 25:2023 lists. Our results show that using only DAST tools is the preferred option for four OWASP Top 10:2021 risk categories while using only SAST tools is preferred for only three risk categories. Either approach is effective for two of the OWASP Top 10:2021 risk categories. For CWE Top 25:2023 list, all three approaches were equally effective and found vulnerabilities belonging to three risk categories each. We also found that none of the tools were able to detect any vulnerability in one OWASP Top 10:2021 risk category and in eight CWE Top 25:2023 categories. This highlights a critical limitation of popular tools. The most effective DAST tool was OWASP Zed Attack Proxy (ZAP), especially for detecting vulnerabilities in broken access control, insecure design, and security misconfiguration risk categories. Yasca was the best-performing SAST tool, and outperformed all other tools at finding high-severity vulnerabilities. For medium-severity and low-severity levels, the DAST tools Iron Web application Advanced Security testing Platform (WASP) and Vega performed better than all the other tools. These findings reveal key insights, such as, the superiority of DAST tools for detecting certain types of vulnerabilities and the indispensability of SAST tools for detecting high-severity issues (due to detailed static code analysis). This study also addresses significant limitations in previous research by testing multiple real-world Web applications across diverse domains (technology, health, and education), enhancing generalization of the findings. Unlike studies that

rely primarily on proprietary tools, our use of open-source SAST and DAST tools ensures better reproducibility and accessibility for organizations with limited budget.

# INTRODUCTION

## Background and motivation

Ensuring the security of software and Web applications is paramount in today's digital landscape. There are numerous incidents reported where exploitation of a well-known vulnerability led to serious consequences. For example, in February 2024, a major data breach was reported in which over 54 million user profiles were exposed due to a misconfiguration that resulted in the compromise of sensitive data (*Hughes, 2024*). In October 2023, a medical diagnostic company, Redcliffe Labs, faced a 7 TB data breach of medical records because no password was being used to protect their database (*Ford, 2023*). According to Verizon's 2023 Data Breach Investigations Report, Web application attacks accounting for nearly 40% of data breaches and high-profile incidents, such as the SolarWinds and Equifax breaches, underscore the need for robust security testing (*Verizon, 2023*).

Despite the severity and scale of these incidents, current Web application security practices remain inadequate primarily due to limited budgets, time constraints, and insufficient training of development and testing teams (*RedEdgeSecurity, 2024*). These constraints often prevent organizations from conducting comprehensive security testing during development or after release. This is obvious from the observation made by *Touseef et al. (2019)* that the most frequently occurring risks are based on injection, cross-site scripting, and sensitive data exposure and these vulnerabilities are not particularly challenging to detect. While automated tools and different approaches exist to streamline the vulnerability assessment process, their effectiveness, scope, and coverage are limited or poorly understood. This is a significant research gap and the need for empirically evaluating the effectiveness and efficiency of available web application security testing approaches was also identified by *Aydos et al. (2022)*. Moreover, it makes sense that the effectiveness of tools and approaches should be assessed by aligning the findings with widely-accepted industry frameworks.

The key novelty of this work lies in its multifaceted assessment of automated security testing tools, which has not been explored comprehensively or with real-world Web applications in prior studies. Our study categorizes the detected vulnerabilities according to the latest industry benchmarks, namely the OWASP Top 10: 2021 and CWE Top 25: 2023 list. This is to ensure actionable insights can be obtained for practitioners through the remediation steps provided by the maintainers of these benchmarks, such as the Web Security Testing Guide (WSTG) (https://github.com/OWASP/wstg). Furthermore, we compare the performance of these tools in detecting vulnerabilities, offering critical

recommendations that can be used for improving security practices and mitigating risks early in the development lifecycle. By bridging this gap, our study contributes to a deeper understanding of security testing approaches and provides a practical framework for enhancing Web application security.

Evaluating multiple security testing approaches and tools is essential because no single tool or method can comprehensively detect vulnerabilities across different stages of the software development lifecycle. Using a static application security testing (SAST) approach allows for early detection and remediation of a broader range of vulnerabilities within the development process, minimizing risks before deployment. Essentially, SAST is a white-box testing method, meaning it requires access to the source code of the application being tested. On the other hand, dynamic application security testing (DAST) approach is a black-box testing method, in which a tool has no access to an application's source code. It examines an application, while it is running, to find vulnerabilities in the same way an actual attacker would. This approach is crucial for identifying vulnerabilities in live applications, where different risks may surface. Together, these two approaches maximize coverage and strengthen security, underscoring the necessity of multifaceted evaluation.

There are two widely-accepted, community-led initiatives that aim to assist developers with security testing. The first one, from MITRE Corporation, is called common weakness enumeration (CWE); it lists hardware and software weaknesses that can become vulnerabilities (*MITRE, 2023*). The CWE database is compiled as a result of extensive research, analysis, and consensus-building among experts in the field (*Dimitrov, 2022*; *MITRE, 2023*). Its top 25 list enumerates vulnerabilities that pose the greatest risk to software integrity and security. The second initiative is called the Open Web Application Security Project (OWASP) and it maintains a top 10 list containing the most critical security risks to Web applications (*OWASP, 2021*). OWASP top 10 is list a community-driven compilation that categorizes vulnerabilities according to their severity and prevalence in Web applications (*OWASP, 2021*). Together, the latest versions of these lists (*i.e.*, the OWASP Top 10:2021 and CWE Top 25:2023), serve as a valuable resource for addressing prevalent Web application security concerns. Tables A1–A3 provide a description of each risk category and links to the official Web page for each risk category for more information.

The relevance and impact of the OWASP Top 10:2021 and CWE Top 25:2023 lists are underscored by their widespread use in multiple research studies, such as *Chaleshtari et al. (2023)*, *Shahid et al. (2022)*, *Li (2020)*, and *Priyawati, Rokhmah & Utomo (2022)*. However, these studies primarily focus on individual tools or specific vulnerability categories and lack a systematic and comparative evaluation of multiple automated tools that utilise SAST and DAST methodologies. Additionally, none of these works have comprehensively mapped detected vulnerabilities to both OWASP Top 10:2021 and CWE Top 25:2023 benchmarks. This leaves a critical research gap where no alignment of empirical findings with widely accepted standards has been made.

Similarly, the prior work by *Khanum, Qadir & Jehan (2023)* laid an important foundation by empirically investigating the effectiveness of OWASP ZAP—a single DAST tool—for detecting OWASP Top 10:2021 vulnerabilities across seventy different Web

applications. While their findings highlighted certain strengths and limitations of OWASP ZAP, the study relied on a single testing tool and it also did not explore vulnerabilities in the context of CWE Top 25:2023 list. Furthermore, this work did not provide a comparative analysis of how multiple tools (both SAST and DAST) perform across diverse vulnerability categories, limiting its generalizability and practical application in broader contexts.

This research addresses the gaps and limitations mentioned above and significantly expands the scope of prior studies by evaluating multiple SAST and DAST tools using a diverse set of real-world Web applications. Unlike previous works, this study adopts a dual-mapping methodology that categorizes findings according to both OWASP Top 10:2021 and CWE Top 25:2023 lists, thereby offering a more comprehensive view of vulnerabilities. This is a novel contribution to the field, as no prior research has systematically assessed the combined effectiveness of SAST and DAST tools, while aligning results with these two widely-used lists. We aim to expand the scope of *Khanum, Qadir & Jehan (2023)* by using multiple SAST and DAST tools, and assess a diverse set of real-world target Web applications. Also, we will map the results to OWASP Top 10:2021 and CWE Top 25:2023 lists. By addressing these gaps, this study will not only improves our understanding of automated security testing tools but also provides actionable insights for developers and security practitioners, ultimately enhancing Web application security practices.

In summary, this work evaluates the effectiveness of SAST and DAST approaches (by utilising multiple SAST and DAST tools) for finding vulnerabilities in real-world Web applications that can be mapped to risk categories included in the OWASP Top 10:2021 and CWE Top 25:2023 lists. The main contributions of this work are:

1. Security testing of seventy-five Web applications using nine popular, free, and open source vulnerability assessment tools and the availability of the results for the research community.

2. Investigation of DAST and SAST approaches (*via* multiple tools) for security testing with respect to finding vulnerabilities that can be mapped to risk categories included in the OWASP Top 10:2021 and CWE Top 25:2023 lists. Our results show that using only DAST approach is recommended for detecting four risk categories from OWASP Top 10:2021 list and three risk categories from CWE Top 25:2023 list. The SAST only approach is best for three OWASP Top 10:2021 risk categories and three CWE Top 25:2023 risk categories. Both approaches were equally effective for two OWASP Top 10:2021 risk categories and three CWE Top 25:2023 risk categories. Our results also identify OWASP ZAP as the performing DAST tool and Yasca as the best performing SAST tool.

3. A detailed validation of the suitability of OWASP ZAP tool for detection of OWASP Top 10:2021 vulnerabilities. Our research shows that OWASP ZAP performs consistently well and is particularly suited to finding vulnerabilities that belong to *A01:2021 Broken Access Control* and *A05:2021 Security Misconfiguration* risk categories.

# RELATED WORK

Multiple studies have assessed the security of existing government or university Web sites using OWASP Top 10:2017 list or the OWASP ZAP tools. Examples range from countries such as Indonesia (*Helmiawan et al., 2020*), Libya (*Murah & Ali, 2018*), Türkiye (*Akgul, 2016*), Pakistan (*Ghazanfar et al., 2021*), and Nigeria (*Idris et al., 2017*). These studies compared the security of live Web sites, based on the count and severity of the vulnerabilities discovered. This is the established method for DAST or post-release security testing.

*Shahid et al. (2022)* compared the effectiveness of eleven commercial and open-source DAST tools (Acunetix, Nessus, Netsparker, Appscan, HP WebInspect, OWASP ZAP, Wapiti, Arachni, Nikto, Burp Suite, and W3af) for Web application security testing. They compared the precision (true positive and false positive rates) for detecting Cross-Site Scripting (XSS) and Structured Query Language injection (SQLi) vulnerabilities and found that the commercial tool Acunetix and the open-source tool OWASP ZAP both achieved 100% precision. These tests were carried out using the well-known Damn Vulnerable Web Application (DVWA), meaning that it is not possible to extrapolate their findings to real-world Web applications.

In *Lachkov, Tawalbeh & Bhatt (2022)*, a testbed environment was used to assess the effectiveness of two DAST tools, Nessus and OpenVAS. The objective of this study was to enhance a company's security posture by securing its network, firewall, servers, clients, and applications. Their findings demonstrated the importance of DAST tools for post-release security testing. Likewise, *Kunda & Alsmadi (2022)* used five open-source DAST tools (OWASP ZAP, SoapUI, Jok3r, SQLMap, and Nikto) to test the security of their custom-built transportation Web application. Although their findings are beneficial, the fact that they used only one target Web application means that it is not possible to generalise the effectiveness of these tools to other Web applications.

In contrast to the DAST approach, the number of recent studies that use only SAST tools is very limited. *Croft et al. (2021)* utilised the SAST approach and the Software Vulnerability Prediction (SVP) method for identifying vulnerabilities in C/C++ open-source projects. They compared three SAST tools, namely Flawfinder, Cppcheck, and RATS with a SVP model. They found that SAST tools have a high false positive rate, are best for supporting manual inspection, and recommended for a reduced code range.

*Li (2020)* is one of the few studies to use the SAST approach and to map identified vulnerabilities to OWASP Top 10:2017 and CWE Top 25:2019 lists. The researchers scanned the source code of a malware detection mobile app using the commercial tool Checkmarx. They also remediated the identified vulnerabilities using the recommendations from Checkmarx. This scanning and remediation process was repeated three times and eventually terminated when only one low severity vulnerability remained. Although this study demonstrated the effectiveness of Checkmarx, it should be noted that Checkmarx is a commercial tool and the use of only one target application makes it hard to generalise the findings.

It is because of these shortcomings (in utilising only SAST or only DAST approach) that more recent studies examine multiple tools that implement different approaches. For example, *Cruz, Almeida & Oliveira, 2023* compared SAST, DAST, and SCA (Software Composition Analysis) approaches for application security using multiple SAST tools (including ESLint, Semgrep, Bandit, Codacy, Deepsource, Flake8, Horusec, Prospector, Radon, and SonarQube) and multiple DAST tools (such as OWASP ZAP, Nikto, Arachni, Beef, Detectify, Golismero, Invicti, Nogotifail, Stackhawk, Vega, Wapiti, and Wfuzz). The comparison was carried out in terms of the programming languages supported, budget requirements, ease of setup, and format of generated report. Based on their findings, the authors recommended OWASP ZAP and Bandit as the best tools. However, they did not specify the vulnerabilities found, their impact, or the target Web applications.

*Tudela et al. (2020)* explored three security testing techniques for Web applications: Interactive Application Security Testing (IAST), SAST, and DAST. They use Contrast and CxIAST tools for IAST, Fortify and FindSecurityBugs tools for SAST, and OWASP ZAP and Arachni for DAST. These tools, a mix of open-source and commercial, were evaluated based on the OWASP Top 10 List using seven metrics (including precision and true positive detection rate). The researchers concluded that IAST tools achieved the best results when combined with DAST tools. One important limitation of this study is that it relied on 320 test cases from the exploitable OWASP Benchmark (https://github.com/OWASP-Benchmark) project instead of real-world Web applications.

*Setiawan, Erlangga & Baskoro (2020)* employed the IAST, SAST, and DAST approaches for analyzing vulnerabilities in Government X's Web sites based on the OWASP Top 10:2017 list using tools like Jenkins, OWASP ZAP, and SonarQube. They identified 81 high-risk vulnerabilities detected using SAST methodology, 94 using IAST methodology, and 13 using DAST approach. Although these results are very informative, the use of Web applications from only one domain is a significant limitation.

In a systematic review, *Alazmi & De Leon (2022)* evaluated 12 out of 30 most popular Web application scanners, in terms of their detection rates and accuracy. The efficacy for detecting OWASP Top 10 vulnerability types was compared for each tool. Their findings showed that SQLi and XSS vulnerability types were the most common and that the other types of vulnerabilities were rarely tested. Also, they reported that Burp Suite Pro exhibits superior performance for detecting XSS vulnerabilities with a detection rate of 88.9% while OWASP ZAP achieved detection rate of only 80%. These findings are a source of motivation for selecting multiple tools that use different approaches and also for validating the finding of each tool across a wide range of Web applications. The focus on a limited number of OWASP Top 10 vulnerabilities can also be noted in *Priyawati, Rokhmah & Utomo (2022)*. These researchers used OWASP ZAP to identify a total of 12 vulnerabilities that belonged to only four OWASP Top 10 risk categories (*A01:2021 Broken Access Control*, *A03:2021 Injection*, *A05:2021 Security Misconfiguration*, and *A08:2021 Software and Data Integrity Failures*).

Table 1 represents a summary of the important studies discussed in this section. It can be easily observed that using multiple real-world Web applications as targets is rare. We address this limitation by using multiple real-world Web applications (from diverse

**Table 1 Comparison of related work.**

| Source | Approach | Tools | Mapping to OWASP or CWE list | Target web app. |
|---|---|---|---|---|
| *Tudela et al. (2020)* | SAST | FindSecurityBugs, Fortify95 | OWASP Top 10:2017 | OWASP Benchmark project |
| | DAST | Arachni, OWASP ZAP | | |
| | IAST | CxIAST | | |
| *Setiawan, Erlangga & Baskoro (2020)* | SAST | SonarQube | OWASP Top 10:2017 | Government X Web sites |
| | DAST | API ZAP | | |
| | IAST | Jenkins | | |
| *Li (2020)* | SAST | Checkmarx | OWASP Top 10:2017 & CWE Top 25:2019 | Mobile malware detection app |
| *Cruz, Almeida & Oliveira (2023)* | SAST | Bandit, Codacy, Deepsource, ESLint, Flake8, Horusec, Prospector, Pylint, Radon, Semgrep, SonarQube | OWASP Top 10:2021 | Not mentioned |
| | DAST | Arachni, Beef, Dtectify, Golismero, Invicti, Nikto, Nogotifail, OWASP ZAP, Stackhawk, Vega, Wapiti, Wfuzz | | |
| | SCA | Back Duck, FOSSA, Npm audit, OWASP D.C, Safety, SourceClear, Steady, Yarn Audit | | |
| *Khanum, Qadir & Jehan (2023)* | DAST | OWASP ZAP | OWASP Top 10:2021 | 70 Web apps |

domains) in order to enhance the generalizability of our results. Also, unlike studies that relied primarily on proprietary tools, we use open-source SAST and DAST tools to improve reproducibility and accessibility for organizations with limited budgets. Additionally, we map the detected vulnerabilities to the latest versions of both the OWASP Top 10:2021 and CWE Top 25:2023 risk categories, which is entirely omitted in earlier studies. The importance of SAST tools should not be underestimated as they are vital for a 'shift left' in the development process where vulnerabilities are found early in the life cycle when they are less costly to mediate (*Dawoud et al., 2024*). These tools can also be used to promote the adoption of Secure Software Development Life Cycle (S-SDLC) and facilitate use of automated testing *via* continuous integration and deployment (CI/CD) pipelines.

## METHODS
### Research questions
The overall objective is to gauge the effectiveness of various Web application security testing approaches and tools in terms of finding OWASP Top 10 and CWE Top 25 vulnerabilities in real-world Web applications. This objective can be achieved by answering the following four research questions:

- RQ1: Which approach, SAST or DAST, is more effective for assessing Web application security (in terms of finding OWASP Top 10:2021 vulnerabilities and CWE Top 25:2023 vulnerabilities)?

- RQ2: Which tool is most effective at assessing Web application security (in terms of finding OWASP Top 10:2021 vulnerabilities and CWE Top 25:2023 vulnerabilities)?
- RQ3: Which tool is most effective at finding high-severity, medium-severity, and low-severity vulnerabilities in Web applications?
- RQ4: Is OWASP ZAP consistently effective in finding vulnerabilities when used to test a large set of real-world Web applications from different domains?

## METHODOLOGY

Figure 1 presents an outline of steps used in this research. Phase-I, conducted earlier by *Khanum, Qadir & Jehan (2023)* focused on finding OWASP Top 10:2021 vulnerabilities in Web applications using the OWASP ZAP tool. Phase-II, which is carried out in this work, aims to find OWASP Top 10:2021 vulnerabilities and CWE Top 25:2023 weaknesses using multiple SAST and DAST tools. Although using multiple tools is generally the recommended approach for vulnerability assessment, the aim is to determine which tool is most suitable for finding which type of vulnerability. This would ensure that tool selection is effective.

Overall, Phase II consists of seven main steps. At the end of these steps, each research question is addressed.

Steps 1, 2, and 3 were applied across all research questions (RQ1, RQ2, and RQ3). Initially, 75 target web applications from three popular domains were selected (Step 1). A set of static application security testing (SAST) tools and dynamic application security testing (DAST) tools were chosen (Step 2), and each web application was deployed locally (Step 3). Each tool was used to scan the web applications (Step 4), generating and saving detailed reports (Step 5). The findings from these reports were then categorized based on the identified vulnerabilities, following the OWASP Top 10: 2021 and CWE Top 25: 2023 standards (Step 6). For RQ4, the categorized data from Step 6 were further analysed in Step 7, where the findings were tabulated to address the all of the four research questions.

This systematic process ensured a clear understanding of vulnerabilities and their classifications to effectively address the research questions. It provides a more comprehensive security evaluation hence validating results from our semi-automated workflow. The details of each step in Phase-II is described below:

1. **Select 4 SAST and 5 DAST Tools**-A total of nine (9) security tools were selected as follows:

   ○ SAST tools: Yasca, Progpilot, Snyk, and SonarQube.
   ○ DAST tools: OWASP ZAP, Wapiti, Vega, Iron WASP, and Burp Suite.

The tools listed above were selected based on their effectiveness, ease of use, and ability to identify vulnerabilities. Tools that were not free (*e.g.*, Acunetix, Netsparker) or restricted to finding only one type of vulnerability (*e.g.*, SQLi, XSSFuzz) were excluded from this research. Table 2 provides a detailed comparison of the selected tools, including features

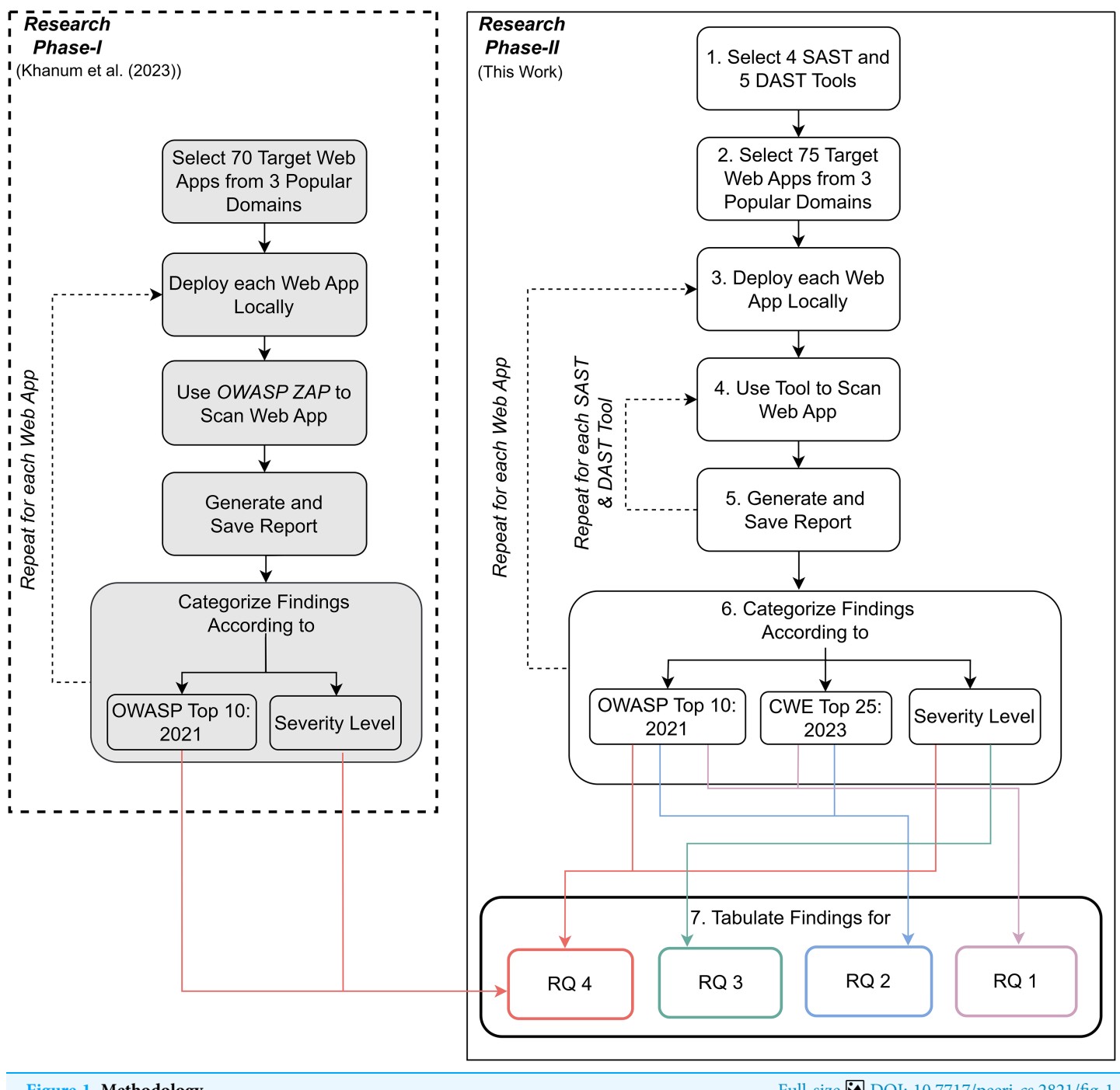

**Figure 1** **Methodology.**

such as the ability to automatically generate a report containing all the findings (see 'Report' column) at the end of a scan. Another useful feature is 'Est. Time'. It represents an estimate of the time taken by the tool to scan a single Web application. For SAST tools, the supported languages are listed in the final column.

**Table 2 Important characteristics of selected tools.**

| No. | Tool | Approach | Interface | Report | Est. time | Language |
|-----|------|----------|-----------|--------|-----------|----------|
| 1 | Yasca | SAST | CLI | Yes | 5 min | PHP, Java, C/C++, Python, JS, Perl, NET |
| 2 | Progpilot | SAST | CLI | No | 15 min | PHP |
| 3 | Snyk | SAST | CLI | No | 10 min | PHP, Java, Golang, Python, JS, Swift, NET |
| 4 | SonarQube | SAST | GUI | Yes | 10 min | PHP, C/C++, Python, C#, NET, Java, JS, Kotlin, Ruby, Swift |
| 5 | OWASP ZAP | DAST | GUI | Yes | 1 h | – |
| 6 | Wapiti | DAST | CLI | Yes | 10 min | – |
| 7 | Vega | DAST | GUI | No | 6 h | – |
| 8 | Iron WASP | DAST | GUI | Yes | 6 h | – |
| 9 | Burp suite | DAST | GUI | Yes | 6 h | – |

Vega, Wapati, OWASP ZAP, and Iron WASP are tools recommended by *Altulaihan, Alismail & Frikha, 2023*. Also, *Albahar, Alansari & Jurcut, 2022* recommended Burp Suite and OWASP ZAP as the best DAST tools for detecting Web application vulnerabilities. These two tools were identified as the most popular tools by *Aydos et al. (2022)*. Additionally, *Amankwah et al. (2022)* showed that Yasca had a high precision rate between 83% and 90.7% for detecting security bugs in Java. In terms of supporting developers' workflow, Snyk reports are very informative and the tool integrates well with GitHub (*i.e.*, new vulnerabilities introduced through pull requests are automatically checked) (*Anupam et al., 2020*). SonarQube, provides excellent automation and generates suggestions for handling vendor branches (*Andrade, 2019*). Lastly, Progpilot is a popular tool endorsed by the OWASP community for detecting vulnerabilities in PHP code (*OWASP, 2024*).

2. **Select 75 target web apps from three popular domains**-A total of seventy-five (75) Web apps from three key domains were selected using convenient sampling. Also, only those Web apps whose source code was available were used. This is a requirement for static analysis. The domains and the number of Web apps selected is as follows:

   ○ Seven Web apps from healthcare domain
   ○ 11 Web apps from education domain
   ○ 57 Web apps from technology domain

These domains were chosen due to their increased vulnerability to cyberattacks, as highlighted in a 2023 report (*WEF, 2023*). To ensure transparency, the names and links of some of these Web apps is provided in Tables A4–A6 (in Appendix A). A complete list is available on GitHub (https://github.com/devNowRO/WebAppSecurity/blob/main/Web apps sources.xlsx).

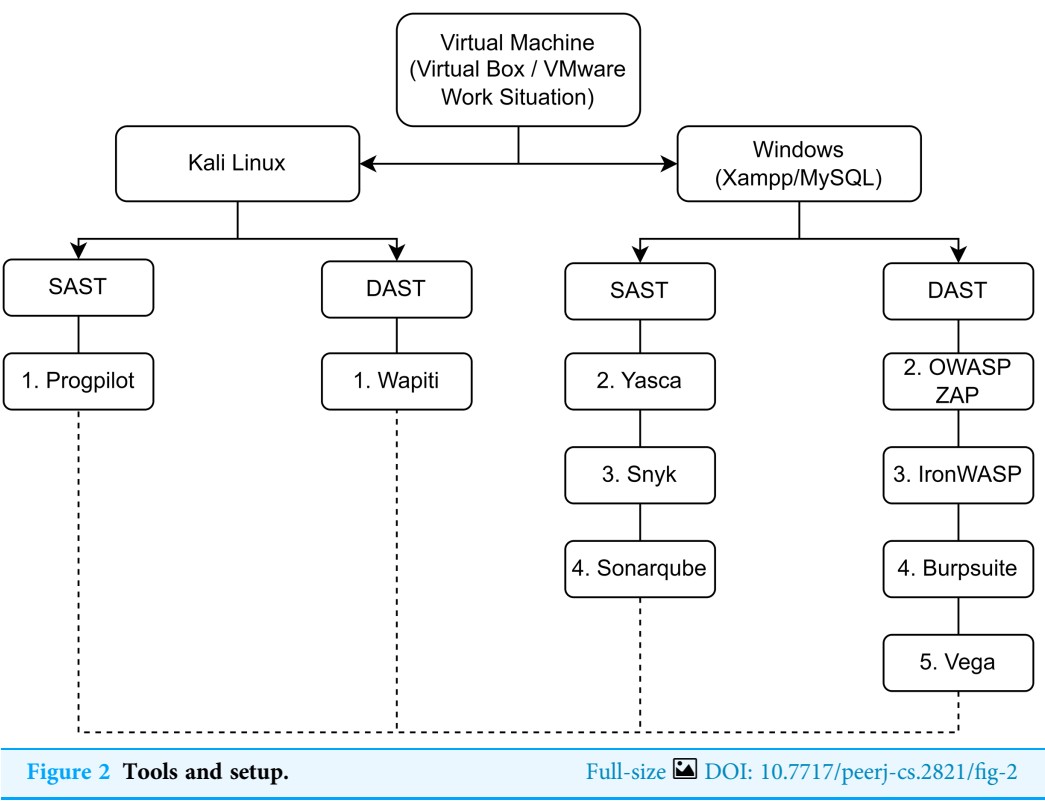

**Figure 2  Tools and setup.**               

3. **Deploy each web app locally**-Each of the 75 Web apps was deployed locally on a Windows operating system environment using the XAMPP Web server and MySQL database.

4. **Use tool to scan web app**-Fig. 2 shows how each tool was deployed including the machine and platform used. The version and other details of each tool is available on GitHub (https://github.com/devNowRO/WebAppSecurity/blob/main/README.md). Essentially, seven tools were deployed on Windows while two tools were deployed on Linux. These two tools, namely Wapiti and Progpilot, are available exclusively on Linux. The use of both operating systems demonstrates the flexibility and adaptability of our testing environment and underscores the versatility of the tools. It also shows that our methodology can be applied in varied environments, which is particularly relevant for diverse organizational setups.

Each target Web app deployed in Step 3 was scanned using all nine tools. For example, the target Web app `Employee Record Management System` was deployed and scanned by Yasca, Progpilot, Snyk, SonarQube, OWASP ZAP, Wapiti, Vega, Iron WASP, and finally Burp Suite. After this, the next target Web app was deployed and scanned (by each of the nine tools) and so on. Each SAST tool used static code analysis to identify vulnerabilities in the source code. Each DAST tool performed runtime analysis to detect vulnerabilities.

5. **Generate and save report**-After each scan, each tool automatically generated detailed vulnerability reports. In some cases, these reports also included suggestions for fixing

the vulnerabilities. The reports were saved and were made publicly available at GitHub (https://github.com/devNowRO/WebAppSecurity/blob/main/README.md).

6. **Categorization of vulnerabilities according to OWASP Top 10:2021, CWE Top 25:2023, and severity level**-The vulnerabilities listed in each report were categorized according to OWASP Top 10:2021 and CWE Top 25:2023 risk categories. Some of the tools (namely Progpilot, OWASP ZAP, Wapiti, and Yasca) provided automatic OWASP and CWE categorizations. For the remaining tools (namely Vega, Iron WASP, Burp Suite, Snyk, and SonarQube), the descriptions of the identified vulnerabilities were manually reviewed and then a suitable category was identified by the authors.

Also, the authors used the information at the Web sites of both these lists to determine mapping of the risk categories. The outcome of this mapping is shown in Table 3 (*MITRE, 2023*; *OWASP, 2021*). For example, it can been seen that the eighth risk category in the OWASP Top 10:2021 list (*A08:2021 Software and Data Integrity Failures*) is mapped to the fifteenth risk category in the CWE Top 25:2023 list, namely *CWE-502: Deserialization of Untrusted Data*. It is also obvious that some risk categories have no equivalent risk categories in the other list. It should be noted that whenever there was a difference in the automatic category identified by the tool and that shown in Table 3, the one provided by the tool was used. This makes sense as the tool is scanning the actual Web app in real-time, while the information provided on the Web site is generic.

Furthermore, all tools (except Wapiti) automatically categorized vulnerabilities in terms of severity (low, medium, high, or critical). For Wapiti, the severity level was identified by the authors with help from the level assigned to similar vulnerabilities by another tool.

As an example, we demonstrate the categorisation process for one sample Web app *i.e.*, `Employee Record Management System`. The reports generated for this Web app by one static analysis tool (*i.e.*, Yasca) and one dynamic analysis tool (*i.e.*, OWASP ZAP) are analysed as described below:

- ○ Analysis of Yasca report-Fig. 3 shows a part of this report that lists the first six vulnerabilities found in the `Employee Record Management System`. As an example, vulnerability #2 is identified as a 'Cross-Site Scripting (XSS)' vulnerability and clicking on its 'Details' button leads to a detailed description of this vulnerability as shown in Fig. 4. The severity label for this vulnerability is provided in column 2 of the report as shown in Fig. 3.

  This vulnerability is mapped to the *A03:2021 Injection* risk category of OWASP Top 10:2021 list and to the *CWE-79 Injection Improper Neutralization of Input During Web Page Generation ('Cross-site Scripting')* risk category of CWE Top 25:2023 list using the information provided on the respective Web pages of these vulnerabilities. The risk category *A03:2021 Injection* in the OWASP Top 10:2021 refers to vulnerabilities where malicious data is sent to an interpreter, leading to attacks like SQL injection. Meanwhile *CWE-79 Injection Improper Neutralization of Input During Web Page Generation ('Cross-site Scripting')* is a specific weakness where missing custom error pages in J2EE (Java™ 2 Platform, Enterprise Edition)

**Table 3** Mapping of OWASP Top 10:2021 to CWE Top 25:2023 risk categories.

| OWASP rank | OWASP Top 10:2021 name of risk category | CWE Top 25:2023 name of risk category | CWE rank |
|---|---|---|---|
| 1 | A01:2021 Broken Access Control | CWE-22: Improper Limitation of a Pathname to a Restricted Directory ('Path Traversal') | 8 |
| | | CWE-276: Incorrect Default Permissions | 25 |
| | | CWE-352: Cross-Site Request Forgery (CSRF) | 9 |
| | | CWE-862: Missing Authorization | 11 |
| | | CWE-863: Incorrect Authorization | 24 |
| 2 | A02:2021 Cryptographic Failures | – | – |
| 3 | A03:2021 Injection | CWE-20: Improper Input Validation | 6 |
| | | CWE-77: Improper Neutralization of Special Elements used in a Command ('Command Injection') | 16 |
| | | CWE-78: Improper Neutralization of Special Elements used in an OS Command ('OS Command Injection') | 5 |
| | | CWE-79: Improper Neutralization of Input During Web Page Generation ('Cross-site Scripting') | 2 |
| | | CWE-89: Improper Neutralization of Special Elements used in an SQL Command ('SQL Injection') | 3 |
| | | CWE-94: Improper Control of Generation of Code ('Code Injection') | 23 |
| 4 | A04:2021 Insecure Design | CWE-269: Improper Privilege Management | 22 |
| | | CWE-434: Unrestricted Upload of File with Dangerous Type | 10 |
| 5 | A05:2021 Security Misconfiguration | – | – |
| 6 | A06:2021 Vulnerable and Outdated Components | – | – |
| 7 | A07:2021 Identification and Authentication Failures | CWE-287: Improper Authentication | 13 |
| | | CWE-306: Missing Authentication for Critical Function | 20 |
| | | CWE-798: Use of Hard-coded Credentials | 18 |
| 8 | A08:2021 Software and Data Integrity Failures | CWE-502: Deserialization of Untrusted Data | 15 |
| 9 | A09:2021 Security Logging and Monitoring Failures | – | – |
| 10 | A10:2021 Server-Side Request Forgery (SSRF) | CWE-918: Server-Side Request Forgery (SSRF) | 19 |
| – | – | CWE-787 Out-of-bounds Write | 1 |
| – | – | CWE-416 Use After Free | 4 |
| – | – | CWE-125: Out-of-bounds Read | 7 |
| – | – | CWE-476: NULL Pointer Dereference | 12 |
| – | – | CWE-190: Integer Overflow or Wraparound | 13 |
| – | – | CWE-119: Improper Restriction of Operations within the Bounds of a Memory Buffer | 17 |
| – | – | CWE-362: Concurrent Execution using Shared Resource with Improper Synchronization ('Race Condition') | 21 |

**Figure 3** Snippet I of Yasca's report for `Employee Record Management System`.

**Figure 4** Snippet II of Yasca's report for `Employee Record Management System`.

**Table 4 Analysis of Yasca's Employee Record Management System.**

| Name of vulnerabilities | OWASP Top 10:2021 category | CWE Top 25:2023 category | Severity level | Number of vulnerabilities |
|---|---|---|---|---|
| Cross Site Scripting | A03:2021 Injection | CWE-79 Improper Neutralization of Input During Web Page Generation ('Cross-site Scripting') | High | 145 |
| SQL Injection | A03:2021 Injection | CWE-89 Improper Neutralization of Special Elements used in an SQL Command ('SQL Injection') | High | 39 |
| Weak Credentials | A07:2021 Identification and Authentication failures | CWE-259 Use of Hard-coded Passwords | Medium | 22 |

## Alert counts by alert type

This table shows the number of alerts of each alert type, together with the alert type's risk level.

(The percentages in brackets represent each count as a percentage, rounded to one decimal place, of the total number of alerts included in this report.)

| Alert type | Risk | Count |
|---|---|---|
| **SQL Injection** | High | 7 (33.3%) |
| **SQL Injection - MySQL** | High | 7 (33.3%) |
| **Absence of Anti-CSRF Tokens** | Medium | 9 (42.9%) |
| **Application Error Disclosure** | Medium | 1 (4.8%) |
| **Content Security Policy (CSP) Header Not Set** | Medium | 12 (57.1%) |
| **Directory Browsing** | Medium | 9 (42.9%) |

**Figure 5 Snippet I of OWASP ZAP's report for Employee Record Management System.**

applications can expose sensitive information. While *A03:2021 Injection* focuses on exploitation, *CWE-79 Injection Improper Neutralization of Input During Web Page Generation ('Cross-site Scripting')* highlights misconfiguration that can aid attackers in gathering intelligence. Once all the vulnerabilities listed in the entire Yasca report were categorised in the same way, the data was tabulated as shown in Table 4. For instance, it can be seen that a total of 145 high-severity vulnerabilities mapped to *A03:2021 Injection* risk category were found. Similarly, a total of 22 medium-severity vulnerabilities mapped to *CWE-798 Use of Hard-coded Credentials* risk category were found.

○ Analysis of OWASP ZAP report-Fig. 5 shows a part of this report that lists the first three vulnerabilities found in the Employee Record Management System. Taking

**4. Risk=Medium, Confidence=Low (2)**

    **1. http://localhost (2)**

        1. **Absence of Anti-CSRF Tokens** (1)

            1. ▼ GET http://localhost/37.Employee-Record-Management-System/erms/registererms.php

| | |
|---|---|
| **Alert tags** | • OWASP_2021_A01<br>• WSTG-v42-SESS-05<br>• OWASP_2017_A05 |
| **Alert description** | No Anti-CSRF tokens were found in a HTML submission form.<br><br>A cross-site request forgery is an attack that involves forcing a victim to send an HTTP request to a target destination without their knowledge or intent in order to perform an action as the victim. The underlying cause is application functionality using predictable URL/form actions in a repeatable way. The nature of the attack is that CSRF exploits the trust that a web site has for a user. By contrast, cross-site scripting (XSS) exploits the trust that a user has for a web site. Like XSS, CSRF attacks are not necessarily cross-site, but they can be. Cross-site request forgery is also known as CSRF, XSRF, one-click attack, session riding, confused deputy, and sea surf.<br><br>CSRF attacks are effective in a number of situations, including:<br><br>* The victim has an active session on the target site. |

**Figure 6** Snippet II of OWASP ZAP's report for `Employee Record Management System`. 

the third vulnerability identified (*i.e.*, 'Absence of Anti-CSRF Token') as an example, we note that scrolling further in the report leads to more details about this vulnerability as shown in Figs. 6 and 7. The severity label for this vulnerability is provided in the 'Risk' column as shown in Fig. 5.

We can see that OWASP ZAP has already mapped this vulnerability to the *A01:2021 Broken Access Control* risk category of OWASP Top 10:2021 list and to the *CWE-352: Cross-Site Request Forgery (CSRF)* risk category of CWE Top 25:2023 list (see Figs. 6 and 7). In general, *A01:2021 Broken Access Control* refers to vulnerabilities where attackers can bypass authorization mechanisms, gaining unauthorized access to data or functionality. Meanwhile *CWE-352: Cross-Site Request Forgery (CSRF)* refers to a specific weakness where an attacker tricks a user into performing unwanted actions on a Web application where they are authenticated. Both these risk categories highlight improper access and user session exploitation.

Once all the vulnerabilities in the entire OWASP ZAP report were categorised in the same way, the data was tabulated as shown in Table 5. For instance, it can be seen that a total of 14 high-severity vulnerabilities mapped to *A01:2021 Broken Access Control* risk category were found. Similarly, a total of 9 medium-severity vulnerabilities mapped to *CWE-548: Broken Access Exposure of Control Information Through Directory Listing* risk category were found.

Finally, a few vulnerabilities reported by OWASP ZAP (and included in Table 5) did not map to a risk category included in the CWE Top 25:2023 list (shown in Table 2).

3. **Absence of Anti-CSRF Tokens**

| | |
|---|---|
| **Source** | raised by a passive scanner (Absence of Anti-CSRF Tokens) |
| **CWE ID** | 352 |
| **WASC ID** | 9 |
| **Reference** | 1. http://projects.webappsec.org/Cross-Site-Request-Forgery<br>2. http://cwe.mitre.org/data/definitions/352.html |

4. **Application Error Disclosure**

| | |
|---|---|
| **Source** | raised by a passive scanner (Application Error Disclosure) |
| **CWE ID** | 200 |
| **WASC ID** | 13 |

5. **Content Security Policy (CSP) Header Not Set**

| | |
|---|---|
| **Source** | raised by a passive scanner (Content Security Policy (CSP) Header Not Set) |
| **CWE ID** | 693 |

**Figure 7 Snippet III of OWASP ZAP's report for** `Employee Record Management System`.

**Table 5 Analysis of OWASP ZAP's report for** `Employee Record Management System`.

| Name of vulnerability | OWASP Top 10:2021 category | CWE Top 25:2023 category | Severity level | Count of vulnerabilities |
|---|---|---|---|---|
| SQL Injection | A01:2021 Broken Access Control | CWE-89 Improper neutralization of Special Elements used in an SQL Command ('SQL Injection') | High | 7 |
| SQL Injection-MYSQL | A01:2021 Broken Access Control | CWE-89 Improper Neutralization of Special Elements used in an SQL Command ('SQL Injection') | High | 7 |
| Absence of Anti-CSRF Tokens | A01:2021 Broken Access Control | CWE-352 Cross-Site Request Forgery (CSRF) | Medium | 9 |
| Application Error Disclosure | A05:2021 Security Misconfiguration | – | Medium | 1 |
| Content Security Policy (CSP) Header Not Set | A05:2021 Security Misconfiguration | – | Medium | 12 |
| Directory Browsing | A01:2021 Broken Access Control | – | Medium | 9 |
| Missing Anti-clickjacking Header | A05:2021 Security MISCONFIGURATION | – | Medium | 8 |
| Parameter Tampering | A04:2021 Insecure Design | – | Medium | 2 |
| Vulnerable JS Library | A06:2021 Vulnerable and Outdated Components | – | Medium | 2 |
| Big Redirect Detected (Potential Sensitive Information Leak) | A04:2021 Insecure Design | – | Low | 3 |
| Cookie no HttpOnly Flag | A05:2021 Security Misconfiguration | – | Low | 2 |
| Cookie without SameSite Attribute | A01:2021 Broken Access Control | – | Low | 2 |

(Continued)

| Table 5 (continued) | | | | |
|---|---|---|---|---|
| Name of vulnerability | OWASP Top 10:2021 category | CWE Top 25:2023 category | Severity level | Count of vulnerabilities |
| Server Leaks Information *via* "X-Powered-By" HTTP Response Header Field(s) | A01:2021 Broken Access Control | – | Low | 13 |
| Server Leaks Version Information *via* "Server" HTTP Response Header Field | A05:2021 Security Misconfiguration | – | Low | 24 |
| X-Content-Type-Options Header Missing | A05:2021 Security Misconfiguration | – | Low | 14 |

For example 'Content Security Policy (CSP) Header Not Set' and 'X-Content-Type-Options Header Missing' vulnerabilities could be mapped to *CWE-693: Protection Mechanism Failure* but this CWE is not included in CWE Top 25:2023 list. Likewise, multiple vulnerabilities listed in Table 5 could be mapped to *CWE-200: Exposure of Sensitive Information to an Unauthorized Actor* risk category but this CWE is also not included in the CWE Top 25:2023 list (shown in Table 2) and therefore it is not mentioned in Table 5. It should be noted here that reporting these vulnerabilities is outside the scope of this work.

The above categorization process was repeated for the reports generated by each of the nine tools (four SAST and five DAST tools) for all seventy-five Web applications. This iterative process ensured the completeness and consistency of the analysis. A detailed guide for replicating this work is available at GitHub (https://github.com/devNowRO/WebAppSecurity/blob/main/Methodology.txt).

7. **Tabulation of Findings for:**

○ **RQ1**-To determine which approach was more effective for assessing Web application security (in terms of finding vulnerabilities belonging to OWASP Top 10:2021 and CWE Top 25:2023 risk categories), the number of Web applications in which vulnerabilities found using 'Only SAST approach', using 'Only DAST', or 'Both Approaches' was tabulated. The findings are presented in Table A7 (in Appendix A) for the OWASP Top 10:2021 list and in Table A8 (in Appendix A) for the CWE Top 25:2023 list.

○ **RQ2**-To determine which tool was most effective for assessing Web application security (in terms of finding vulnerabilities belonging to OWASP Top 10:2021 and CWE Top 25:2023 risk categories), the number of vulnerabilities found (belonging to each risk category) was tabulated. For the OWASP Top 10:2021 list, the findings are presented in Table A9 (in Appendix A). For the CWE Top 25:2023 list, the findings are shown in Tables A10 and A11 (in Appendix A).

○ **RQ3**-To determine which tool was most effective at finding vulnerabilities at each severity level, the number of vulnerabilities found at each severity level (high, medium, and low) was tabulated. The findings are presented in the **Results for RQ3** section.

○ **RQ4**-To determine if OWASP ZAP is consistently effective in finding vulnerabilities, the number Web applications in which vulnerabilities belonging to OWASP Top 10:2021 list was tabulated along with the severity level of each vulnerability. Only the reports generated by scanning each of the 75 Web applications using OWASP ZAP were used. The emphasis on OWASP ZAP is because it is recommended by the OWASP project and warrants additional evaluation. Therefore, a methodological evaluation of OWASP ZAP's consistency and accuracy using target Web applications from distinct domains lays the groundwork for understanding how OWASP ZAP performs under varying scopes and conditions. The findings are presented in Table A12 (in Appendix A) together with the findings from Phase-I (*Khanum, Qadir & Jehan, 2023*).

## RESULTS

### Results for RQ1

The findings for SAST and DAST approaches, categorized according to OWASP Top 10:2021, are presented in Fig. 8. The X-axis represents the OWASP Top 10:2021 risk categories, while the Y-axis displays the number of Web applications in which vulnerabilities from each category were identified. There are three bars for each risk category, namely 'Only SAST' approach, 'Only DAST' approach, and 'Both Approaches'. For instance, in Fig. 8, vulnerabilities belonging to *A02:2021: Cryptographic Failures* risk category:

- were found by the 'Only SAST' approach in 14 target Web applications. These vulnerabilities were not identified by any DAST tool.
- were found by the 'Only DAST' approach in three target Web applications. These vulnerabilities were not identified by any SAST tool.
- were found by 'Both Approaches' in one target Web application. In other words, these vulnerabilities were by SAST and by DAST approach.

This distinction highlights the contribution of each approach and the combined detection capability of both approaches in cases where both approaches were able to detect vulnerabilities within the same risk category.

First, it is obvious that for four risk categories, utilising only DAST tools is more effective than utilising only SAST tools. These categories are *A01:2021 Broken Access Control, A04:2021 Insecure Design, A06:2021 Vulnerable and Outdated Components* and *A08:2021 Software and Data Integrity Failures*.

Conversely, using only SAST tools is more effective (than using only DAST tools) for the *A02:2021 Cryptographic Failure, A07:2021 Identification and Authentication Failures* and *A-10:2021 Server-Side Request Forgery* category.

However, both approaches are effective for *A03:2021 Injection* and *A05:2021 Security Misconfiguration* categories. None of the tools, using either approach, were able to find vulnerabilities belonging to *A09:2021 Security Logging and Monitoring Failures* risk

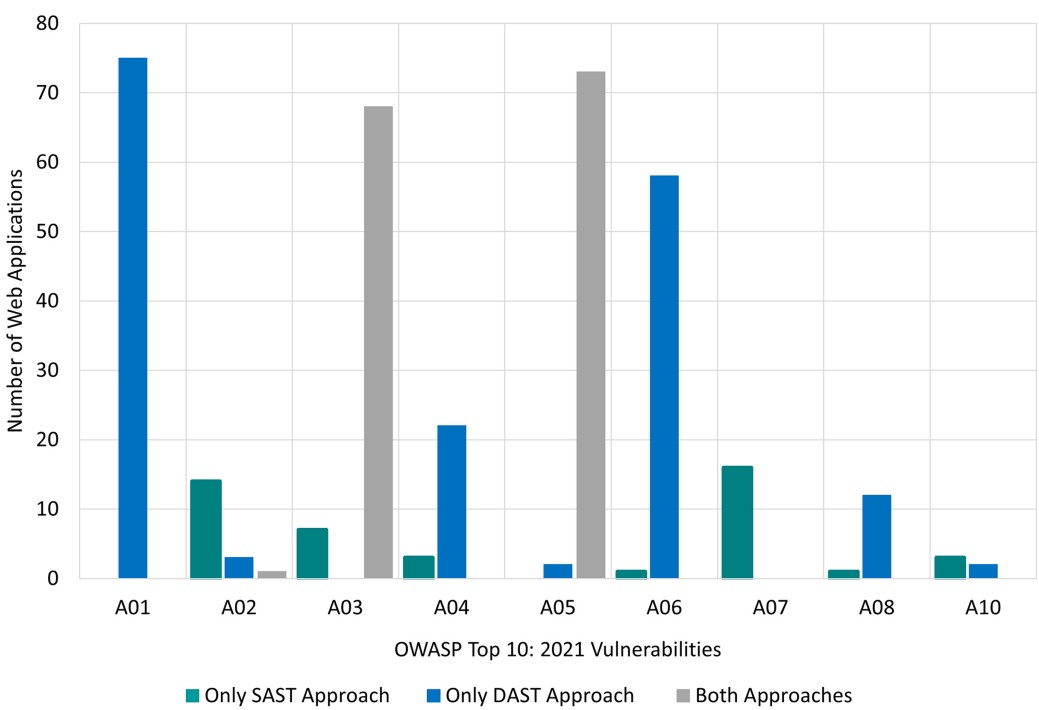

**Figure 8** RQ1: OWASP-based comparison of SAST and DAST approaches.

category. For clarity, this category was excluded from Fig. 8. Table A7 (in Appendix A) summarises the findings for all 10 risk categories included in the OWASP Top 10:2021 list.

The findings, with respect to CWE Top 25:2023, are presented in Fig. 9, where the X-axis represents the CWE Top 25:2023 risk categories, and the Y-axis shows the number of Web applications in which each vulnerability was identified. Similar to Fig. 8, there are three bars for each risk category, namely 'Only SAST' approach, 'Only DAST' approach, and 'Both Approaches'.

To start with, utilising only SAST tools is more effective than using only DAST tools for more than half of the CWEs shown in Fig. 9. For three of the CWEs (*CWE-352 Cross-Site Request Forgery (CSRF)*, *CWE-862 Missing Authorization*, *CWE-119 Improper Restriction of Operations within the Bounds of a Memory Buffer*), using only DAST approach found the most vulnerabilities. In contrast, using only SAST approach was able to identify the most vulnerabilities belonging to CWEs (*CWE-287 Improper Authentication*, *CWE-798 Use of Hard-coded Credentials* and *CWE-306 Missing Authentication for Critical Function*). Both approaches were able to identify *CWE-79 Improper Neutralization of Input During Web Page Generation ('Cross-site Scripting')*, *CWE-89 Improper Neutralization of Special Elements used in an SQL Command ('SQL Injection')*, and *CWE-22 Improper Limitation of a Pathname to a Restricted Directory ('Path Traversal')*.

Unfortunately, none of the tools (regardless of approach) were successful in finding vulnerabilities belonging to eight risk categories. This can be seen in Table A8

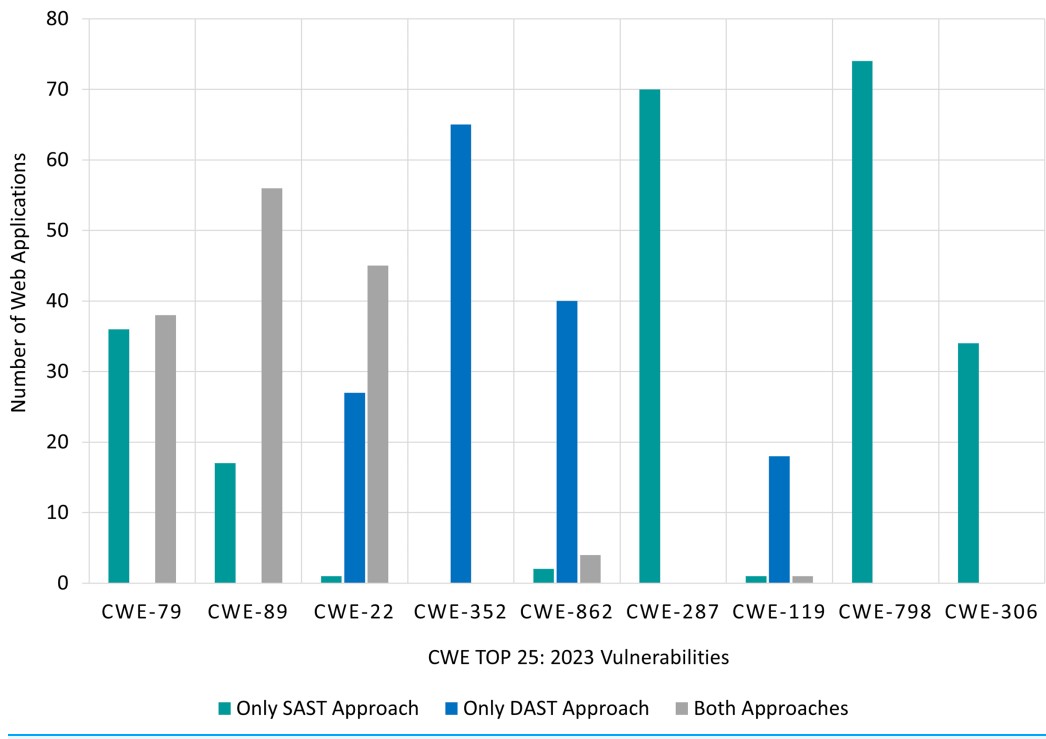

**Figure 9** RQ1: CWE-based comparison of SAST and DAST approaches.

(in Appendix A) that presents a summary of the findings for all 25 risk categories. For the sake of clarity, the above-mentioned eight risk categories (along with eight other risk categories that were found in very few Web applications) are excluded from Fig. 9.

In summary, the best approach for developers is shown in Table 6. This recommendation is based on the approach that detected vulnerabilities (belonging to each risk category) in the highest number of Web applications. For instance, to identify *A01:2021 Broken Access Control* vulnerability, the 'Only DAST' approach is recommended as it was able to identify this vulnerability in all 75 Web applications (see Fig. 8 and Table A7). Likewise, 'Only SAST' approach was able to identify *CWE-798: Use of Hard-coded Credentials* in 74 out of 75 Web applications (see Fig. 9 and Table A8).

## Results for RQ2

The findings for the nine tools categorised according to OWASP Top 10:2021 is presented in Fig. 10. The X-axis represents the OWASP Top 10:2021 risk categories and the Y-axis indicates the number of vulnerabilities identified. For each category, nine bars are shown; one for each tool. To aid clarity, only six of the most commonly detected categories are included in Fig. 10. This also helps ensure a more focused analysis on the effectiveness of each tool in identifying the most significant vulnerabilities. The results for all ten categories in presented in Table A9.

Two tools stand out in terms of the number of found vulnerabilities. Firstly, Yasca dominated the identification of vulnerabilities belonging to the *A03:2021 Injection*

**Table 6 RQ1: recommended approach.**

| Approach | OWASP Top 10:2021 | CWE Top 25:2023 |
|---|---|---|
| Only DAST | A01:2021 Broken Access Control | CWE-119: Buffer Overflow |
| | A04:2021 Insecure Design | CWE-352: Cross-Site Request Forgery (CSRF) |
| | A06:2021 Vulnerable and Outdated Components | CWE-862: Missing Authorization |
| | A08:2021 Software and Data Integrity Failures | |
| Only SAST | A02:2021 Cryptographic Failure | CWE-287: Improper Authentication |
| | A07:2021 Identification and Authentication Failures | CWE-798: Use of Hard-coded Credentials |
| | A10:2021 Server-Side Request Forgery (SSRF) | CWE-306: Missing Authentication for Critical Function |
| Both | A03:2021 Injection | CWE-79: Improper Neutralization of Input During Web Page Generation ('Cross-site Scripting') |
| | A05:2021 Security Misconfiguration | CWE-89: SQL Injection |
| | | CWE-22: Path Traversal |

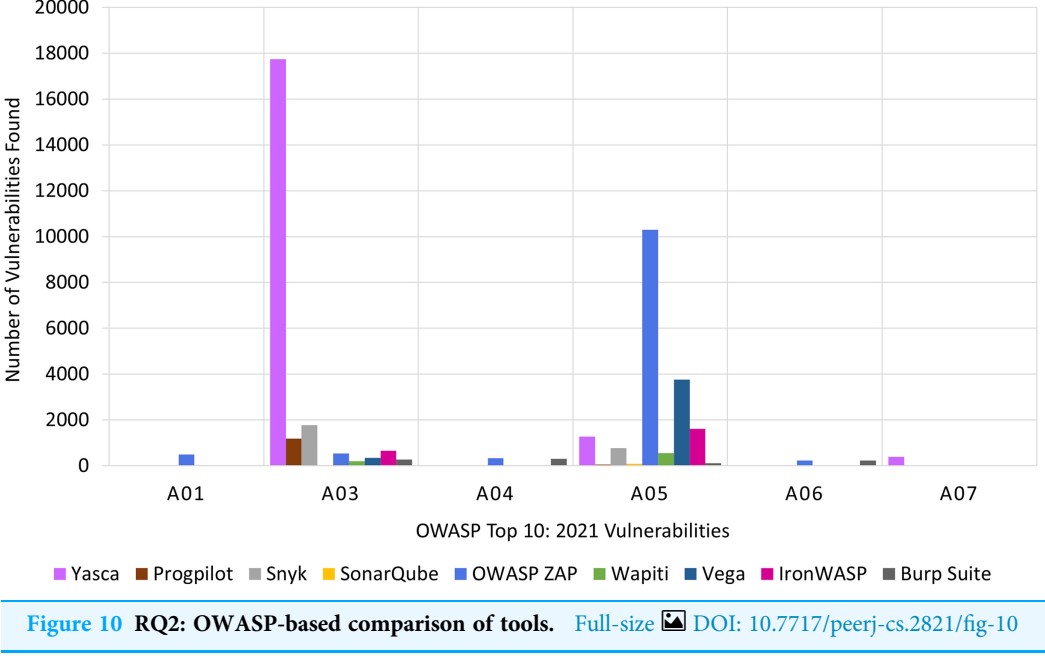

**Figure 10 RQ2: OWASP-based comparison of tools.**

category while OWASP ZAP dominated in the *A05:2021 Security Misconfiguration* categories by finding the most vulnerabilities.

In the case of CWE Top 25:2023, Fig. 11 presents the number of vulnerabilities detected by each tool. The X-axis represents the CWE Top 25:2023 risk categories, while the Y-axis shows the number of vulnerabilities found across all the target Web applications. For each category, nine bars are shown; one for each tool.

In Fig. 11, we focused only on the six most frequently detected categories, as other categories had a very small number of vulnerabilities that could not be effectively shown in the figure. This allows for a more focused comparison of the tools' performance in detecting the most prevalent vulnerabilities within the CWE Top 25:2023 list. The results for all twenty-five categories in presented in Tables A10 and A11.

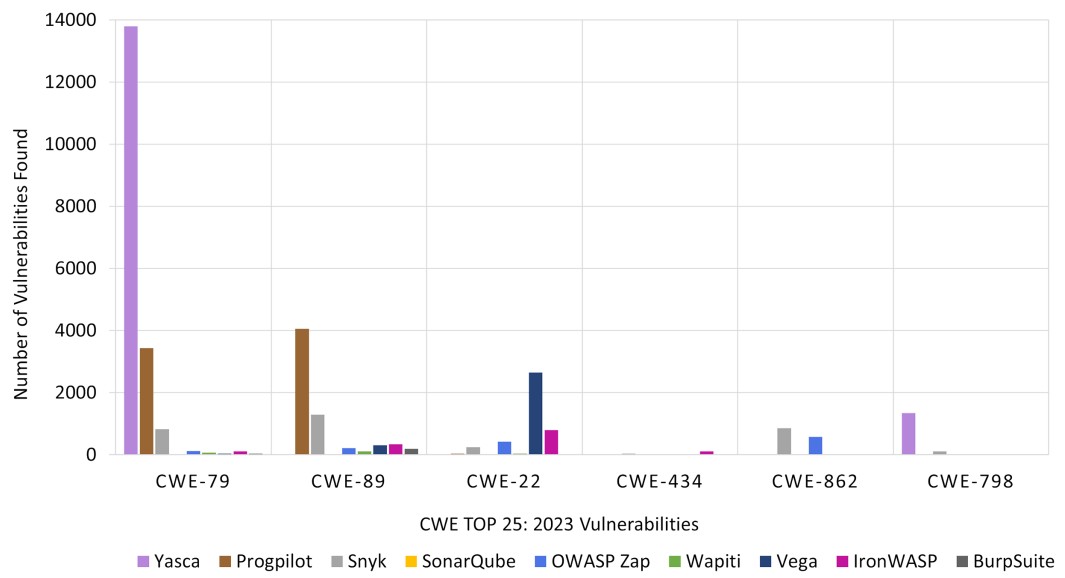

**Figure 11** RQ2: CWE-based comparison of tools.

Unmistakably, Yasca was the dominant tool for *CWE-79: Improper Neutralization of Input During Web Page Generation*. Also, for *CWE-798 Use of Hard-coded Credentials* category, Yasca found more vulnerabilities than any other tool even though the count was less than 2,000.

Progpilot performed better than all the other tools for *CWE-89: SQL Injection*. It was also the second-best for finding vulnerabilities in the *CWE-79: Improper Neutralization of Input During Web Page Generation* category.

For *CWE-22: Path Traversal*, Vega outperformed all the other tools. Finally, for the remaining two categories, Synk found the most vulnerabilities in *CWE-862: Missing Authorization* and Iron WASP had the highest count for *CWE-434: Unrestricted File Upload*.

In a nutshell, the most appropriate tool for developers for each OWASP Top 10:2021 risk category and for each CWE Top 25:2023 category is shown in Table 7. For example, OWASP ZAP is the most effective tool for three of the OWASP Top 10:2021 risk categories, namely *A01:2021 Broken Access Control*, *A04:2021 Insecure Design*, and *A05:2021 Security Misconfiguration*. Similarly, Yasca is the best tool for finding vulnerabilities that belong to *CWE-79: Improper Neutralization of Input During Web Page Generation ('Cross-site Scripting')*. Yasca is also best at finding vulnerabilities in the *A03:2021 Injection* and *A07:2021 Identification and Authentication Failures)* risk categories.

## Results for RQ3

Table 8 presents the number of high-severity, medium-severity, and low-severity vulnerabilities detected by each tool. Yasca has the highest count of 19,465 for

**Table 7  RQ2: recommended tool.**

| Tool | OWASP Top 10:2021 | CWE Top 25:2023 |
|------|-------------------|-----------------|
| Yasca | A03:2021 Injection | CWE-79: Improper Neutralization of Input During |
|  | A07:2021 Identification and Authentication Failures | Web Page Generation ('Cross-site Scripting') |
|  |  | CWE-798: Use of Hard-coded Credentials |
| Progpilot | – | CWE-89: SQL Injection |
| Snyk | – | CWE-862: Missing Authorization |
| SonarQube | – |  |
| OWASP ZAP | A01:2021 Broken Access Control | – |
|  | A04:2021 Insecure Design |  |
|  | A05:2021 Security Misconfiguration |  |
| Wapiti | – |  |
| Vega | – | CWE-22: Path Traversal |
| Iron WASP | – | CWE-434: Unrestricted File Upload |
| Burp suite | A06:2021 Vulnerable and Outdated Components |  |

**Table 8  RQ3: findings for severity level (number of vulnerabilities found).** The highest number of vulnerabilities at each level of severity are shown in bold.

| Severity | Yasca | Progpilot | Synk | SonarQube | Tool Vega | Iron WASP | Burp Suite | Wapiti | OWASP ZAP |
|----------|-------|-----------|------|-----------|-----------|-----------|------------|--------|-----------|
| High | **19,465** | 4,001 | 1,776 | 36 | 845 | 676 | 392 | 291 | 217 |
| Medium | 6 | 0 | 402 | 10 | 501 | **1,471** | 6 | 0 | 690 |
| Low | 342 | 30 | 687 | 32 | **2,900** | 143 | 530 | 507 | 340 |

high-severity vulnerabilities. The second best tool is Progpilot, with a considerably lower count of 4,001 high-severity vulnerabilities. Interestingly, both are SAST tools.

In the case of medium-severity vulnerabilities, Iron WASP recorded the highest count of 1,471. This is almost twice the number found by OWASP ZAP. It should also be noted here that Progpilot and Wapiti were not able to find any medium-severity vulnerabilities.

For low-severity vulnerabilities, Vega had the highest count of 2,900 vulnerabilities. In contrast, the performance of SonarQube was poor although this could be because the free version used in the research and it is likely to be less effective than the paid version.

A few other observations can be made from Table 8. Firstly, reputable tools such as OWASP ZAP and Burp Suite did not outperform other tools as expected. OWASP ZAP had the second highest count for medium-severity vulnerabilities while Burp Suite secured the third highest count for low-severity vulnerabilities. This comparatively poorer performance of Burp Suite could be because the free community edition was used in this study. Secondly, the tool with the largest range of performance was Progpilot. It had the second highest count for high-severity vulnerabilities but performed extremely poorly for

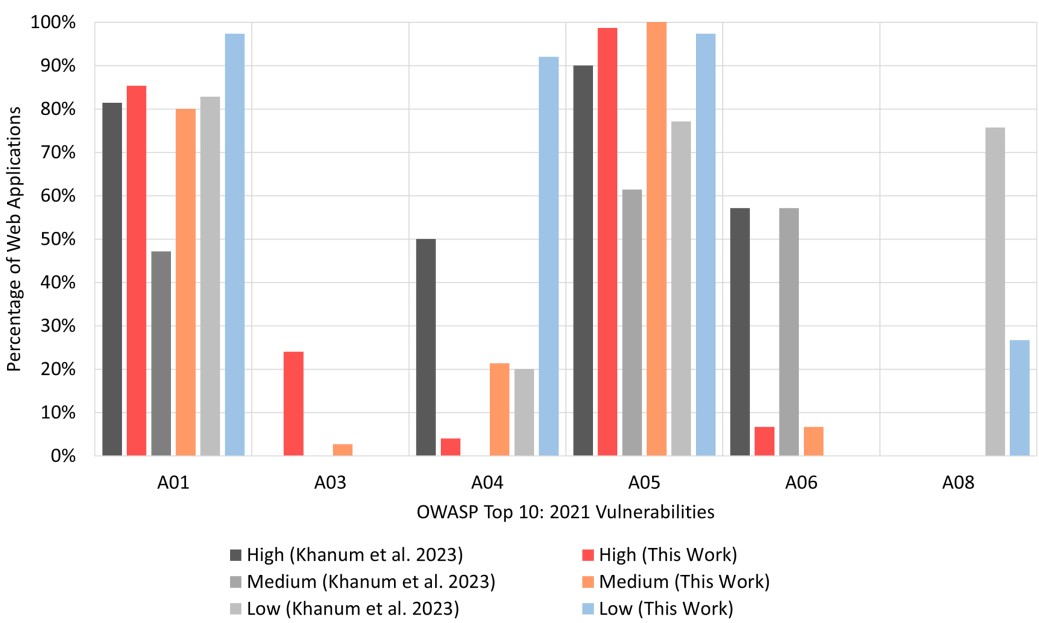

**Figure 12 RQ4: comparison of OWASP ZAP's performance (based on % of web applications).**

medium- and low-severity vulnerabilities. This could be explained by the fact that it is a language-specific (PHP) static analysis tool.

Overall, Yasca is the best choice for finding high-severity vulnerabilities. Iron WASP seems promising for medium-severity vulnerabilities, while Vega was found to be effective for low-severity vulnerabilities.

### Results for RQ4

In RQ4, we sought to check that the effectiveness of OWASP ZAP, as presented in *Khanum, Qadir & Jehan (2023)*, also applies when a lager set of real-world applications are used as targets. It is important to note that the target Web applications and domains examined in *Khanum, Qadir & Jehan (2023)*, including banking, e-commerce, and recruitment portals differ from those analysed in this work (namely healthcare, education, and technology). The complete list of of target Web applications and their OWASP ZAP reports, used in *Khanum, Qadir & Jehan (2023)*, are available on GitHub (https://github.com/devNowRO/WebAppSecurity/tree/main/Khanum-et-al-2023).

The comparison, illustrated in Fig. 12, uses percentages instead of counts to ensure fairness because *Khanum, Qadir & Jehan (2023)* used 70 Web applications as targets, while this study used 75. The X-axis represents the OWASP Top 10:2021 risk category and the Y-axis shows the percentage of Web applications in which vulnerabilities were identified. For clarity, we have focused on only six most frequently occurring 10 OWASP Top 10:2021 risk categories in Fig. 12. The reader is referred to Table A12 (in Appendix A) for a detailed comparison of the findings from *Khanum, Qadir & Jehan (2023)* and this study.

The results clearly confirm the consistent effectiveness of OWASP ZAP in identifying vulnerabilities across a wide range of Web application domains. In particular, the

performance for *A01:2021 Broken Access Control* and *A05:2021 Security Misconfiguration* risk categories is very similar. *A01:2021 Broken Access Control* pertains to cases where users may act beyond their intended permissions, leading to unauthorized access to sensitive information, while *A05:2021 Security Misconfiguration* includes instances of improper configurations that leave systems vulnerable to attack.

For the remaining four risk categories shown in Fig. 12, there is a slightly more noticeable difference in performance. The most notable deviation in performance is observed in the *A03:2021 Injection* risk category. In *Khanum, Qadir & Jehan (2023)*, OWASP ZAP did not detect any instances of this vulnerability, whereas in this study 20% of the Web applications exhibited vulnerabilities in this category. Also, there is considerable difference in severity levels for these four risk categories. For instance, low-severity vulnerabilities belonging to *A04:2021 Insecure Design* risk category, were found in about 20% of the Web applications in *Khanum, Qadir & Jehan (2023)* but in more than 90% of the Web applications in this work. It should also be noted that our results also confirm that OWASP ZAP struggles to detect vulnerabilities in certain risk categories. This is discussed in greater detail in the **Discussion of Results for RQ4** section.

## DISCUSSION

In this work, we set out to determine the best approach and the best free tool for finding vulnerabilities in each OWASP Top 10:2021 and CWE Top 25:2023 risk categories. Our results are compared with the findings reported in existing work in Table 9. Firstly, most other studies rely on commercial tools to find OWASP vulnerabilities in one or two well-known vulnerable Web Applications. Secondly, our work has two more unique features. To begin with, we used both OWASP Top 10:2021 and CWE Top 25:2023 lists. The only other research that does this was *Li (2020)* but it used previous versions of these lists and deployed only one custom-made app. This makes it impossible to make any meaningful comparison of their results with ours. The next unique aspect of our work is that we scanned multiple real-world Web applications using nine different tools and made our results publicly available for further research.

### Discussion of results for RQ1

Based on our findings, we recommend approaches for several risk categories in Table 6. For instance, 'Only DAST' approach is recommended for four OWASP Top 10:2021 risk categories and three CWE Top 25:2023 risk categories. Similarly, the results also helped us recommend the 'Only SAST' approach for three OWASP Top 10:2021 risk categories and three CWE Top 25:2023 risk categories (see Table 6 for the names of these risk categories).

Only in the case of following two OWASP Top 10:2021 risk categories and three CWE Top 25:2023 risk categories were either approach equally effective.

- *A03:2021 Injection*
- *A05:2021 Security Misconfiguration*

**Table 9 Comparison of findings.**

| Source | OWASP Top 10 | CWE Top 25 | Real-world apps | Finding(s)/Contribution(s) | Limitation(s) |
|---|---|---|---|---|---|
| *Tudela et al. (2020)* | 2017 | N | N | Combination of SAST (Fortify); DAST(Arachni, OWASP ZAP); IAST (CCE) approach is best. | OWASP Benchmark Project |
| *Setiawan, Erlangga & Baskoro (2020)* | 2017 | N | Y | IAST (Jenkins, API ZAP and SonarQube) approach provides greater test accuracy. | 1 domain |
| *Li (2020)* | 2017 | 2019 | Y | Checkmarx for SAST | 1 custom-made app |
| *Cruz, Almeida & Oliveira (2023)* | 2021 | N | – | OWASP ZAP for DAST Bandit for SAST | No information about target apps |
| *Khanum, Qadir & Jehan (2023)* | 2021 | N | Y[1] | OWASP ZAP is effective for five categories | Only OWASP ZAP |
| This work | 2021 | 2023 | Y[2] | 1. DAST approach is suitable for OWASP Top 10:2021 and using SAST approach is suitable for CWE Top 25:2023 | Tools did not identify vulnerabilities in all risk categories. |
| | | | | 2. OWASP ZAP is the best tool for OWASP Top 10:2021 and Yasca and Synk are the best for CWE Top 25:2023 | |
| | | | | 3. Yasca is best for high severity, Iron WASP for medium severity, and Vega for low severity vulnerabilities. | |
| | | | | 4. OWASP ZAP is consistent in effectiveness in terms of count and severity of vulnerabilities | |

**Notes:**
[1] 50 live; 20 Locally-hosted.
[2] 75 Locally-hosted.

- *CWE-79: Improper Neutralization of Input During Web Page Generation ('Cross-site Scripting')*
- *CWE-89: Improper Neutralization of Special Elements used in an SQL Command ('SQL Injection')*
- *CWE-22: Improper Limitation of a Pathname to a Restricted Directory ('Path Traversal')*

In such cases, the selection of approach can be made based on non-technical criteria such as developer experience, ease of integration into workflow, or budget constraints.

It is interesting to note that our results confirm the recommendation of *Tudela et al. (2020)* and *Cruz, Almeida & Oliveira, 2023* that combining the two approaches is the most effective strategy. However, we go one step further by:

- identifying the risk categories that can best be detected using 'Only SAST' or 'Only DAST' approach. For example, from Table A7, we can see that 'Only SAST' is best for *A07:2021 Identification and Authentication Failures* risk category ('Only DAST' approach was not able to find any vulnerability belonging to this category). Likewise, we can note that for *A01:2021 Broken Access Control*, 'Only SAST' approach was not effective but 'Only DAST' was extremely effective. Similarly, for the CWE Top 25:2023 list, we found that *CWE-352: Cross-Site Request Forgery (CSRF)* is best identified using 'Only DAST' and *CWE-287: Improper Authentication* is best detected using 'Only SAST'.

- identifying the risk categories that were not detected by either approach (see the rows with 0's in Tables A7 and A8). The names of these risk categories (one from OWASP Top 10:2021 list and eight from CWE Top 25:2023 list) are given below:

- *A09:2021 Security Logging and Monitoring Failures*
- *CWE-787 Out-of-bounds Write*
- *CWE-416 Use After Free*
- *CWE-125 Out-of-bounds Read*
- *CWE-476 NULL Pointer Dereference*
- *CWE-190 Integer Overflow or Wraparound*
- *CWE-502 Deserialization of Untrusted Data*
- *CWE-863 Incorrect Authorization*
- *CWE-276 Incorrect Default Permissions*

This limitation is not unique to our work and has been documented by *Alazmi & De Leon (2022)*, *Priyawati, Rokhmah & Utomo (2022)*, and *Khanum, Qadir & Jehan (2023)*. There could be several reasons for the absence of vulnerabilities belonging to certain risk categories:

- The Web application was developed using the Secure Software Development Life Cycle (S-SDLC) meaning that it incorporated secure coding practices such as input validation, output encoding, and robust authentication. These measures would effectively prevent common vulnerabilities like 'Insecure Key Management', 'Credential Stuffing' Vulnerabilities, 'Insufficient Logging' and 'Insecure URL Handling'. Additionally, a type-safe, memory-safe programming language could have been selected for development that was not susceptible to the missing vulnerabilities and included smooth error handling. For example, languages like Java, Perl, and TypeScript perform their own memory management and therefore are unlikely to contain vulnerabilities such as *CWE-787 Out-of-bounds Write*, *CWE-416 Use After Free*, or *CWE-476 NULL Pointer Dereference*. Likewise, if the programming language or data constructs are selected carefully, it is also possible to avoid vulnerabilities that belong to *CWE-190 Integer Overflow or Wraparound*.
- The Web application's operational flow during scanning did not include code, data, or interaction that would expose certain vulnerabilities. For example, the absence of functionalities like file uploads or sensitive data handling reduces the attack surface, limiting opportunities for exposure of vulnerabilities that may be categorised as belonging to *A10:2021 Server-Side Request Forgery (SSRF)*.
- Automated vulnerability assessment tools may also have failed to detect certain issues, resulting in false negatives. An example of this would be vulnerabilities that belong to *A09:2021 Security Logging and Monitoring Failures* as deficiencies in logging and monitoring are best determined by expert reviews or after an incident.

- Lastly, the limitation of false positives and false negatives is recognised as an issue in automated Web application security testing tools (*Aydos et al., 2022*).

## Discussion of results for RQ2

With regard to the best tool (see Table 7), we find that our recommendation of OWASP ZAP (a DAST tool) is similar to that of *Tudela et al. (2020)* and *Cruz, Almeida & Oliveira (2023)*. Our recommendation is based on its comparatively superior performance for *A01:2021 Broken Access Control*, *A04:2021 Insecure Design*, *A05:2021 Security Misconfiguration* risk categories compared to the eight other tools. Likewise, for *A03:2021 Injection* and *A07:2021 Identification and Authentication Failures* risk categories, we recommend the SAST tool Yasca. Finally, for *A06:2021 Vulnerable and Outdated Components* risk category, Burp Suite is recommended. No recommendation is made for the remaining three OWASP Top 10:2021 risk categories because the number of vulnerabilities found were very negligible or zero (refer to Table A9). The reader is referred to the **Discussion of Results for RQ1** subsection for possible reasons why no vulnerabilities (belonging to certain categories) were detected.

For the CWE Top 25:2023 list, recommendations are made for only six risk categories in Table 7. The complete results for CWE Top 25:2023 list are presented in Table A10) and Table A11. Yasca proved to be the only tool capable of finding vulnerabilities that belong to more than one risk category *i.e., CWE-79: Improper Neutralization of Input During Web Page Generation ('Cross-site Scripting')* and *CWE-798: Use of Hard-coded Credentials*. Progpilot, Synk, Vega, and Iron WASP are recommended for one risk category each.

It should also be noted that our data showed Yasca, Progpilot, and Synk as the most effective SAST tools which is at variance with the SAST tools mentioned by *Li (2020)*, *Cruz, Almeida & Oliveira, 2023*, and *Tudela et al. (2020)*. This could be because we only considered free versions of SAST tools and the fact that SAST tools are language-dependent. These reasons could also explain the comparatively poorer performance of well-known tools like Burp Suite in our study-it recommended for only one risk category *A06:2021 Vulnerable and Outdated Components* in Table 7.

Overall, we recommend Yasca for SAST analysis and OWASP ZAP for dynamic analysis based on the number of vulnerabilities detected by all tools.

## Discussion of results for RQ3

There are very few studies that assess tools according to the severity-level of vulnerabilities they detect. Our results show that a SAST tool, namely Yasca, is the best choice for finding high-severity vulnerabilities (refer to Table 8). On the other hand, DAST tools perform comparatively better for medium-severity and low-severity vulnerability detection.

However, it should be remembered that SAST tools are known for high false-positive rates (*Croft et al., 2021*). Their use is best complemented by manual inspection and should be limited to certain sections of code. This means that Yasca should be considered for

finding high-severity vulnerabilities but with the caveat that false-positives should be manually examined. If the source code of a Web application is not available, then DAST tools like Vega are recommended for high-severity and low-severity levels while Iron WASP is recommended for medium-severity levels (refer to Table 8).

## Discussion of results for RQ4

The results of this study, with respect to checking the efficacy of OWASP ZAP on a more diverse set of target Web applications, overlaps to some extent with the findings reported by *Priyawati, Rokhmah & Utomo (2022)* and *Alazmi & De Leon (2022)*. The former suggested that the most frequently found categories by OWASP ZAP are *A01:2021 Broken Access Control*, *A04:2021 Insecure Design*, *A05:2021 Security Misconfiguration*, and *A08:2021 Software and Data Integrity Failures*. All four of these risk categories are included in Fig. 12 thereby demonstrating the consistency of OWASP ZAP's performance. The latter study, *Alazmi & De Leon (2022)* noted that most effective tools tend to centre on the detection of SQLi and XSS attacks which fall in the *A03 Injection* category. In our findings, less than a quarter of the Web applications had vulnerabilities belonging to this category while in *Khanum, Qadir & Jehan (2023)* this was zero. This difference could be explained by the use of both live and locally deployed Web applications in *Khanum, Qadir & Jehan (2023)*. It is possible that injection risks are not adequately addressed in small deployments whereas larger organizations with live Web applications may have dedicated teams that have effectively mitigate such vulnerabilities. Another influencing factor could be the selection of Web application domains. *Khanum, Qadir & Jehan (2023)* primarily focused on e-commerce, banking, and recruitment domains, which tend to have comparatively stronger security measures than the three domains used in this study (*i.e.*, education, healthcare, and technology).

Lastly, it can be seen from Table A9 that OWASP ZAP fails to detect any vulnerability belonging to the following categories:

- *A02:2021 Cryptographic Failures*-this risk category refers to the use of default cryptography keys or the use of weak cryptography keys. Such vulnerabilities are best detected with SAST analysis instead of DAST analysis tools such as OWASP ZAP. Table A9 confirms this by showing that only SAST analysis tools, such as Synk, were able to detect vulnerabilities belonging to this category.

- *A07:2021 Identification and Authentication Failures*-vulnerabilities belonging to this risk category were only detected by the SAST tool Yasca (see Table A9). None of the other tools, including OWASP ZAP was able to detect any vulnerabilities belonging to this category. This category typically includes vulnerabilities such as the use of hard-coded passwords and credentials. Therefore, it is very likely that only a comprehensive SAST tool, like Yasca, that supports languages that other SAST tools do not support (such as C/C++ and Perl) is comparatively more effective.

- *A09:2021 Security Logging and Monitoring Failures*-this is a unique risk category that can be challenging to test. From Table A9, it can be seen that none of the eight tools was able to identify any vulnerabilities that could be mapped to it. According to OWASP, the

vulnerabilities belonging to this risk category are determined using interviews or asking if attacks were detected during a penetration test. Therefore, our results are excellent for highlighting the importance of non-technical methods and recommending their use to complement technical security testing.

- *A10:2021 Server-Side Request Forgery (SSRF)*-this is a new risk category added to the latest version of OWASP Top 10 list and it occurs when a remote resource is fetched without validating the user-supplied URL. Overall, it is known to have a low incidence rate and only Synk and Iron WASP were able to detect a few vulnerabilities belonging to this risk category (Table A9). OWASP ZAP was not able to detect this risk category in this work or in *Khanum, Qadir & Jehan (2023)* (see Table A12). This limitation could be taken into consideration during plans for further development of OWASP ZAP.

The exploitation of the above risk categories can have severe consequences, especially the second-highest ranking *A02:2021 Cryptographic Failures*, but their detection seems to require use of techniques beyond OWASP ZAP's scope. This underscore the importance of supplementary, approaches, tools or manual review for thorough Web security assessment.

## Limitations and threats to validity

Firstly, this study relied on locally-hosted target Web applications to maintain consistent testing conditions. It is likely that the findings may differ (to some extent) if the security assessment was repeated using updated versions of these Web applications or in a live production environment with different server configurations, databases, or network settings. Specifically, the technical constraint of this work include hosting applications in a local environment using XAMPP and MySQL, which may not fully replicate the complexities of production environments. Key factors such as distributed databases, diverse attack surfaces, different input data, and increased traffic loads in real-world scenarios could influence the identification of vulnerabilities. To account for such changes, future research should focus on testing live applications within real-world environments. Also, as the threat landscape is rapidly evolving, future work should use the latest risk categories, such as the updated version of CWE Top 25 released in 2024 (https://cwe.mitre. org/data/definitions/1430.html) and the next version of OWASP Top 10 that is expected to be announced in the first half of 2025 (https://owasp.org/www-project-top-ten/).

Moreover, there are two external threats pertaining to the obtained results. First, the results obtained are specific to the examples studied. To determine if these findings can be generalized to other contexts, additional live applications across various domains should be tested using hybrid testing approaches. Second, each tool is limited in its abilities. For instance, SAST tools like Yasca are limited to finding code vulnerabilities and do not capture actual application behaviour or usage patterns. DAST tools, on the other hand, do not analyse underlying source code and cannot detect vulnerabilities due to insecure coding patterns. Furthermore, any future updates to testing tools may produce inconsistencies in results.

Finally, construct validity could be affected by the evaluation criteria applied to each tool. Variations in vulnerability definitions and manual categorization may lead to

inconsistencies. Additionally, limitations in scan thoroughness, attributed to time constraints, may have affected the depth of vulnerability detection, thereby restricting the comprehensiveness of the findings. Furthermore, this study offers observational insights rather than a causal analysis of tool effectiveness, suggesting that larger sample sizes may be necessary to validate these trends.

## CONCLUSION

This study evaluates seventy-five Web applications using nine different SAST and DAST tools to assess their effectiveness for identifying vulnerabilities belonging to OWASP Top 10:2021 and CWE Top 25:2023 risk categories.

Our findings show that using 'Only DAST' approach is recommended for detecting four risk categories of OWASP Top 10:2021 while using 'Only SAST' approach is recommended for three risk categories. Either approach can be used for *A03:2021 Injection* and *A05:2021 Security Misconfiguration* risk categories. For CWE Top 25:2023, the results are more evenly distributed. Specifically, three different risk categories are best detected by using 'Only DAST', three by 'Only SAST', and also three by 'Both Approaches'.

The best performing DAST tool was OWASP ZAP while Yasca was the best performing SAST tool. Yasca was also able to find the highest number of high-severity vulnerabilities. For medium-severity and low-severity levels, the DAST tools, Iron WASP and Vega, were able to find the most vulnerabilities. Furthermore, tools such as SonarQube (free version) and Wapiti could not be recommended for detecting any risk categories. Burp Suite (free version) was only considered a good option for one OWASP Top 10:2021 risk category. It is hoped that these recommendations can be helpful for developers during tool selection.

Lastly, we showed that the performance of OWASP ZAP is mostly consistent with our earlier work *Khanum, Qadir & Jehan (2023)* in that it is a very effective tool for finding vulnerabilities when tested on large set of real-world Web applications. Specifically, the results for *A01:2021 Broken Access Control* and *A05:2021 Security Misconfiguration* risk categories were very similar.

The main limitation of this work is that none of the tools used were able to detect vulnerabilities in a few risk categories. For the OWASP Top 10:2021 list, no vulnerabilities were detected for the *A09:2021 Security Logging and Monitoring Failures* risk category. According to OWASP's Web site, vulnerabilities in this risk category are determined using interviews conducted during incident response or penetration testing to check if attacks were detected (https://owasp.org/Top10/A09_2021-Security_Logging_and_Monitoring_Failures). This emphasises the importance of including non-technical methods to ensure thorough security testing. In the case of the eight CWE Top 25:2023 risk categories that were not detected (*CWE-787: Out-of-bounds Write*, *CWE-476: NULL Pointer Dereference*, *CWE-416: Use After Free*, *CWE-190: Integer Overflow or Wraparound*, *CWE-863: Incorrect Authorization*, *CWE-502: Deserialization of Untrusted Data*, *CWE-276: Incorrect Default Permissions*, *CWE-125: Out-of-bounds Read*), it is important to note that there could be multiple reasons for this outcome. For example, it is possible to avoid more than half of

these CWEs just by carefully selecting a safe, robust programming language and avoiding risky programming constructs. Future work could explore a wider selection of tools and more realistic deployment of Web applications to ensure thorough automated security testing.

# APPENDIX A

**Table A1  OWASP top 10:2021 list (*OWASP, 2021*).**

| Rank | OWASP top 10:2021 | Brief description |
|------|-------------------|-------------------|
| 1 | A01:2021 Broken Access Control | This occurs when restrictions on what authenticated users are allowed to do are not properly enforced, allowing unauthorized access to sensitive data or functionality. |
| 2 | A02:2021 Cryptographic Failures | This includes weaknesses in cryptographic algorithms, key management, and data protection methods. |
| 3 | A03:2021 Injection | This occurs when untrusted data is sent to an interpreter as part of a command or query, leading to unintended execution of malicious commands (*e.g.*, SQL injection). |
| 4 | A04:2021 Insecure Design | This refers to security flaws that arise from inadequate or missing security controls during the design phase of an application. |
| 5 | A05:2021 Security Misconfiguration | This occurs when security settings are not properly configured, leaving the application vulnerable to attacks. |
| 6 | A06:2021 Vulnerable and Outdated Components | This involves using components (*e.g.*, libraries, frameworks) with known vulnerabilities or outdated versions that are no longer supported. |
| 7 | A07:2021 Identification and Authentication Failures | This covers problems such as flawed authentication mechanisms and inadequate session management. |
| 8 | A08:2021 Software and Data Integrity Failures | This occurs when software or data is tampered with, leading to unauthorized modifications or execution of malicious code. |
| 9 | A09:2021 Security Logging and Monitoring Failures | This pertains to insufficient logging and monitoring, which can hinder detection of security breaches. |
| 10 | A10:2021 Server-Side Request Forgery (SSRF) | This involves attackers inducing a server to make unauthorized requests to internal or external resources. |

**Table A2  CWE Top 25:2023 List-I (*MITRE, 2023*).**

| Rank | CWE Top 25:2023 | Brief description |
|------|-----------------|-------------------|
| 1 | CWE-787: Out-of-bounds Write | This occurs when data is written past the end or before the beginning of the intended buffer, potentially leading to memory corruption or arbitrary code execution. |
| 2 | CWE-79: Improper Neutralization of Input During Web Page Generation ('Cross-site Scripting') | This vulnerability allows attackers to inject malicious scripts into web pages viewed by other users, leading to session hijacking, defacement, or data theft. |
| 3 | CWE-89: Improper Neutralization of Special Elements used in an SQL Command ('SQL Injection') | This occurs when untrusted input is included in SQL queries without proper sanitization, allowing attackers to manipulate or extract database data. |
| 4 | CWE-416: Use After Free | This occurs when a program continues to use a pointer after the memory it references has been freed, potentially leading to crashes or code execution. |
| 5 | CWE-78: Improper Neutralization of Special Elements used in an OS Command ('OS Command Injection') | This allows attackers to execute arbitrary operating system commands by injecting malicious input into a command string. |
| 6 | CWE-20: Improper Input Validation | This occurs when input data is not properly validated, allowing attackers to submit malicious input that can disrupt the application or exploit other vulnerabilities. |

(Continued)

| Rank | CWE Top 25:2023 | Brief description |
|---|---|---|
| 7 | CWE-125: Out-of-bounds Read | This occurs when data is read from memory outside the bounds of the intended buffer, potentially leading to information disclosure or crashes. |
| 8 | CWE-22: Improper Limitation of a Pathname to a Restricted Directory ('Path Traversal') | This allows attackers to access files or directories outside the intended directory, potentially leading to unauthorized data access or system compromise. |
| 9 | CWE-352: Cross-Site Request Forgery (CSRF) | This occurs when an attacker forces a user to execute unwanted actions on a web application in which they are authenticated. |
| 10 | CWE-434: Unrestricted Upload of File with Dangerous Type | This allows attackers to upload malicious files to a server, potentially leading to code execution or system compromise. |
| 11 | CWE-862: Missing Authorization | This occurs when an application does not properly enforce access controls, allowing unauthorized users to perform privileged actions. |
| 12 | CWE-476: NULL Pointer Dereference | This occurs when a program dereferences a pointer that is expected to be valid but is actually NULL, leading to crashes or undefined behavior. |
| 13 | CWE-287: Improper Authentication | This occurs when authentication mechanisms are weak or improperly implemented, allowing attackers to bypass authentication or impersonate users. |

**Table A3 CWE Top 25:2023 list-II (*MITRE, 2023*).**

| Rank | CWE Top 25:2023 | Brief description |
|---|---|---|
| 14 | CWE-190: Integer Overflow or Wraparound | This occurs when an integer operation results in a value that is too large or too small to be represented, potentially leading to unexpected behavior or vulnerabilities. |
| 15 | CWE-502: Deserialization of Untrusted Data | This occurs when untrusted data is deserialized, potentially leading to arbitrary code execution or other malicious outcomes. |
| 16 | CWE-77: Improper Neutralization of Special Elements used in a Command ('Command Injection') | This allows attackers to inject malicious commands into a system command, leading to arbitrary command execution. |
| 17 | CWE-119: Improper Restriction of Operations within the Bounds of a Memory Buffer | This occurs when operations on a memory buffer exceed its bounds, potentially leading to memory corruption or code execution. |
| 18 | CWE-798: Use of Hard-coded Credentials | This occurs when credentials (*e.g.*, passwords or keys) are hard-coded into the application, making them easily discoverable by attackers. |
| 19 | CWE-918: Server-Side Request Forgery (SSRF) | This occurs when an attacker can induce a server to make unauthorized requests to internal or external resources. |
| 20 | CWE-306: Missing Authentication for CriticalFunction | This occurs when a critical function does not require authentication, allowing unauthorized users to perform sensitive actions. |
| 21 | CWE-362: Concurrent Execution using Shared Resource with Improper Synchronization ('Race Condition') | This occurs when multiple threads or processes access a shared resource without proper synchronization, potentially leading to unexpected behavior or vulnerabilities. |
| 22 | CWE-269: Improper Privilege Management | This occurs when privileges are not properly managed, allowing users to gain unauthorized access to sensitive functions or data. |
| 23 | CWE-94: Improper Control of Generation of Code ('Code Injection') | This occurs when an application dynamically generates code without proper validation, allowing attackers to inject malicious code. |
| 24 | CWE-863: Incorrect Authorization | This occurs when an application incorrectly enforces authorization, allowing unauthorized users to access restricted resources. |
| 25 | CWE-276: Incorrect Default Permissions | This occurs when default permissions are set incorrectly, potentially allowing unauthorized access to files or resources. |

**Table A4  Target web applications from healthcare domain.**

| No. | Name of app and URL |
| --- | --- |
| 1 | Blood Bank and Donor |
| 2 | COVID19 Testing Management System |
| 3 | Doctor Appointment Management SyStem |
| 4 | Hospital Management System |
| 5 | Online Birth Certificate System |
| 6 | BP Monitoring Management System |
| 7 | Online Nurse Hiring System |

**Table A5  Target web applications from education domain.**

| No. | Name of app and URL |
| --- | --- |
| 1 | Student Management System |
| 2 | Online Course Registration |
| 3 | Student Result Management System |
| 4 | Student Study Center |
| 5 | Teacher Subject Allocation Management System |
| 6 | Teachers Record Management System |
| 7 | Online Library Management System |
| 8 | Pre-school Enrollment System |
| 9 | Online Education Institutes Managing System |
| 10 | Quiz Web Application |
| 11 | Online Examination System for MCQ |

**Table A6  40 target web applications from technology domain[1].**

| No. | Name of app and URL |
| --- | --- |
| 1 | HOTEL Management |
| 2 | Online Pizza Ordering System |
| 3 | Online Computer and Laptop Store |
| 4 | Quality Beauty Parlour Management System |
| 5 | Book Management System |
| 6 | Online Eyewear Shop Application |
| 7 | Computer Service Management System |
| 8 | Art Gallery MS |
| 9 | Auto Taxi Stand Management System Project |
| 10 | Apartment Visitors Management System |
| 11 | Bank Locker Managament System |
| 12 | Bus Pass Management System |
| 13 | Car Rental Portal |
| 14 | Client Management System |

(Continued)

| No. | Name of app and URL |
|---|---|
| **Table A6** (*continued*) | |
| 15 | Company Visitors Management System |
| 16 | Complaint Management system |
| 17 | Cyber Cafe Management System |
| 18 | Daily Expense Tracker |
| 19 | Dairy Farm Shop Management System |
| 20 | Directory Management System |
| 21 | e-Diary Management System |
| 22 | Employee Leave Management System |
| 23 | Employee Record Management System |
| 24 | GYM Management System |
| 25 | Hostel Management System |
| 26 | Maid Hiring Management System |
| 27 | Men Salon Management System |
| 28 | News Portal Project |
| 29 | Old Age Home Management system |
| 30 | Online Banquet Booking System |
| 31 | online DJ Booking Management System |
| 32 | Online Shopping Portal Project |
| 33 | Online Fire Reporting System |
| 34 | Online Security Guard Hiring System |
| 35 | Rail Pass Management System |
| 36 | Restaurant Table Booking System |
| 37 | Tourism Management System |
| 38 | Zoo Management System |
| 39 | Simple Real-time Chatbox |
| 40 | Online Food Ordering System |

Note:
[1] Complete list at: https://github.com/devNowRO/WebAppSecurity/blob/main/Web%20apps%20sources.xlsx

**Table A7 RQ1: findings for OWASP top 10:2021 risk categories (number of web applications).**

| OWASP category | Only SAST tools | Only DAST tools | Both approaches |
|---|---|---|---|
| A01:2021 Broken Access Control | 0 | 75 | 0 |
| A02:2021 Cryptographic Failures | 14 | 3 | 1 |
| A03:2021 Injection | 7 | 0 | 68 |
| A04:2021 Insecure Design | 3 | 22 | 0 |
| A05:2021 Security Misconfiguration | 0 | 2 | 73 |
| A06:2021 Vulnerable and Outdated Components | 1 | 58 | 0 |
| A07:2021 Identification and Authentication Failures | 16 | 0 | 0 |
| A08:2021 Software and Data Integrity Failures | 1 | 12 | 0 |
| A09:2021 Security Logging and Monitoring Failures | 0 | 0 | 0 |
| A10:2021 Server-Side Request Forgery (SSRF) | 3 | 2 | 0 |

**Table A8 RQ1: findings for CWE top 25:2023 risk categories (number of web applications).**

| CWE category | Only SAST tools | Only DAST tools | Both approaches |
|---|---|---|---|
| CWE-787: Out-of-bounds Write | 0 | 0 | 0 |
| CWE-79: Improper Neutralization of Input During Web Page Generation ('Cross-site Scripting') | 36 | 0 | 38 |
| CWE-89: Improper Neutralization of Special Elements used in an SQL Command ('SQL Injection') | 17 | 0 | 56 |
| CWE-416: Use After Free | 0 | 0 | 0 |
| CWE-78: Improper Neutralization of Special Elements used in an OS Command ('OS Command Injection') | 0 | 2 | 0 |
| CWE-20: Improper Input Validation | 17 | 0 | 0 |
| CWE-125: Out-of-bounds Read | 0 | 0 | 0 |
| CWE-22: Improper Limitation of a Pathname to a Restricted Directory ('Path Traversal') | 1 | 27 | 45 |
| CWE-352: Cross-Site Request Forgery (CSRF) | 0 | 65 | 0 |
| CWE-434: Unrestricted Upload of File with Dangerous Type | 12 | 0 | 0 |
| CWE-862: Missing Authorization | 2 | 40 | 4 |
| CWE-476: NULL Pointer Dereference | 0 | 0 | 0 |
| CWE-287: Improper Authentication | 70 | 0 | 0 |
| CWE-190: Integer Overflow or Wraparound | 0 | 0 | 0 |
| CWE-502: De-serialization of Untrusted Data | 0 | 0 | 0 |
| CWE-77: Improper Neutralization of Special Elements used in a Command ('Command Injection') | 13 | 0 | 0 |
| CWE-119: Improper Restriction of Operations within the Bounds of a Memory Buffer | 1 | 18 | 1 |
| CWE-798: Use of Hard-coded Credentials | 74 | 0 | 0 |
| CWE-918: Server-Side Request Forgery (SSRF) | 4 | 2 | 0 |
| CWE-306: Missing Authentication for Critical Function | 34 | 0 | 0 |
| CWE-362: Concurrent Execution using Shared Resource with Improper Synchronization ('Race Condition') | 11 | 0 | 0 |
| CWE-269: Improper Privilege Management | 2 | 0 | 0 |
| CWE-94: Improper Control of Generation of Code ('Code Injection') | 4 | 3 | 0 |
| CWE-863: Incorrect Authorization | 0 | 0 | 0 |
| CWE-276: Incorrect Default Permissions | 0 | 0 | 0 |

**Table A9 RQ2: findings for OWASP top 10:2021 risk categories (number of vulnerabilities found).** The highest number of vulnerabilities found by each tool are shown in bold.

| Category | Yasca | Prog-pilot | Synk | Sonar-Qube | OWASP ZAP | Wapiti | Vega | Iron-WASP | Burp-suite |
|---|---|---|---|---|---|---|---|---|---|
| A01:2021 Broken Access Control | 0 | 0 | 0 | 0 | 500 | 0 | 0 | 0 | 0 |
| A02:2021 Cryptographic Failures | 1 | 0 | 95 | 1 | 0 | 0 | 0 | 0 | 0 |
| A03:2021 Injection | **17,748** | **1,179** | **1,776** | 2 | 537 | 193 | 349 | 662 | **276** |
| A04:2021 Insecure Design | 0 | 0 | 0 | 0 | 335 | 1 | 0 | 0 | 304 |
| A05:2021 Security Misconfiguration | 1,279 | 50 | 780 | **80** | **10,294** | **547** | **3,761** | **1,611** | 112 |

(Continued)

| Category | Yasca | Prog-pilot | Synk | Sonar-Qube | OWASP ZAP | Wapiti | Vega | Iron-WASP | Burp-suite |
|---|---|---|---|---|---|---|---|---|---|
| A06:2021 Vulnerable and Outdated Components | 0 | 0 | 0 | 1 | 227 | 0 | 0 | 0 | 233 |
| A07:2021 Identification and Authentication Failures | 387 | 0 | 0 | 0 | 0 | 0 | 0 | 0 | 0 |
| A08:2021 Software and Data Integrity Failures | 0 | 0 | 1 | 0 | 28 | 0 | 0 | 0 | 0 |
| A09:2021 Security Logging and Monitoring Failures | 0 | 0 | 0 | 0 | 0 | 0 | 0 | 0 | 0 |
| A10:2021 Server-Side Request Forgery (SSRF) | 0 | 0 | 13 | 0 | 0 | 0 | 0 | 14 | 0 |

**Table A10 RQ2: findings for CWE top 25:2023 risk categories-I (number of vulnerabilities found).** The highest number of vulnerabilities found by each tool are shown in bold.

| Category | Yasca | Prog-pilot | Synk | Sonar-Qube | OWASP ZAP | Wapiti | Vega | Iron-WASP | Burp-Suite |
|---|---|---|---|---|---|---|---|---|---|
| CWE-787: Out-of-bounds Write | 0 | 0 | 0 | 0 | 0 | 0 | 0 | 0 | 0 |
| CWE-79: Improper Neutralization of Input During Web Page Generation ('Cross-site Scripting') | **13,797** | 3,436 | 823 | 0 | 122 | 68 | 39 | 113 | 44 |
| CWE-89: Improper Neutralization of Special Elements used in an SQL Command ('SQL Injection') | 21 | **4,056** | **1,291** | 0 | 219 | **110** | 307 | 336 | **200** |
| CWE-416: Use After Free | 0 | 0 | 0 | 0 | 0 | 0 | 0 | 0 | 0 |
| CWE-78: Improper Neutralization of Special Elements used in an OS Command ('OS Command Injection') | 0 | 0 | 0 | 0 | 4 | 0 | 0 | 37 | 0 |
| CWE-20: Improper Input Validation | 0 | 0 | 59 | 0 | 0 | 0 | 0 | 0 | 0 |
| CWE-125: Out-of-bounds Read | 0 | 0 | 0 | 0 | 0 | 0 | 0 | 0 | 0 |
| CWE-22: Improper Limitation of a Pathname to a Restricted Directory ('Path Traversal') | 0 | 36 | 246 | 0 | 421 | 43 | **2,652** | 792 | 0 |
| CWE-352: Cross-Site Request Forgery (CSRF) | 0 | 0 | 0 | 0 | **9,725** | 0 | 0 | 0 | 0 |
| CWE-434:Unrestricted Upload of File with Dangerous Type | 0 | 0 | 43 | 0 | 0 | 0 | 0 | 109 | 0 |
| CWE-862: Missing Authorization | 0 | 0 | 862 | 0 | 578 | 0 | 0 | 0 | 0 |
| CWE-476: NULL Pointer Dereference | 0 | 0 | 0 | 0 | 0 | 0 | 0 | 0 | 0 |
| CWE-287: Improper Authentication | 0 | 0 | 458 | 30 | 0 | 0 | 0 | 0 | 0 |

**Table A11 RQ2: findings for CWE top 25:2023 risk categories-II (number of vulnerabilities found).** The highest number of vulnerabilities found by each tool are shown in bold.

| Category | Yasca | Prog-pilot | Synk | Sonar-Qube | OWASP ZAP | Wapiti | Vega | Iron-WASP | Burp-Suite |
|---|---|---|---|---|---|---|---|---|---|
| CWE-190: Integer Overflow or Wraparound | 0 | 0 | 0 | 0 | 0 | 0 | 0 | 0 | 0 |
| CWE-502: De-serialization of Untrusted Data | 0 | 0 | 0 | 0 | 0 | 0 | 0 | 0 | 0 |
| CWE-77: Improper Neutralization of Special Elements used in a Command ('Command Injection') | 207 | 0 | 6 | 0 | 0 | 0 | 0 | 0 | 0 |
| CWE-119: Improper Restriction of Operations within the Bounds of a Memory Buffer | 0 | 4 | 0 | 0 | 38 | 0 | 0 | 0 | 0 |
| CWE-798: Use of Hard-coded Credentials | 1,348 | 0 | 117 | 0 | 0 | 0 | 0 | 0 | 0 |
| CWE-918: Server-Side Request Forgery (SSRF) | 0 | 0 | 36 | 0 | 0 | 0 | 0 | 16 | 0 |

| Category | Yasca | Prog-pilot | Synk | Sonar-Qube | OWASP ZAP | Wapiti | Vega | Iron-WASP | Burp-Suite |
|---|---|---|---|---|---|---|---|---|---|
| CWE-306: Missing Authentication for Critical Function | 0 | 0 | 89 | **45** | 0 | 0 | 0 | 0 | 0 |
| CWE-362: Concurrent Execution using Shared Resource with Improper Synchronization ('Race Condition') | 97 | 0 | 0 | 0 | 0 | 0 | 0 | 0 | 0 |
| CWE-269: Improper Privilege Management | 0 | 0 | 3 | 0 | 0 | 0 | 0 | 0 | 0 |
| CWE-94: Improper Control of Generation of Code ('Code Injection') | 0 | 0 | 8 | 0 | 0 | 0 | 0 | 20 | 0 |
| CWE-863: Incorrect Authorization | 0 | 0 | 0 | 0 | 0 | 0 | 0 | 0 | 0 |
| CWE-276: Incorrect Default Permissions | 0 | 0 | 0 | 0 | 0 | 0 | 0 | 0 | 0 |

**Table A12** RQ4: findings for comparison of OWASP ZAP's performance (number of web applications).

| OWASP Top 10:2021 category | Severity level | Khanum, Qadir & Jehan (2023) | This work |
|---|---|---|---|
| A01:2021 Broken Access Control | High | 57 | 64 |
| | Medium | 33 | 60 |
| | Low | 58 | 73 |
| A03:2021 Injection | High | 0 | 18 |
| | Medium | 0 | 2 |
| | Low | 0 | 0 |
| A04:2021 Insecure Design | High | 35 | 3 |
| | Medium | 0 | 16 |
| | Low | 14 | 69 |
| A05:2021 Security Misconfiguration | High | 63 | 74 |
| | Medium | 46 | 75 |
| | Low | 54 | 73 |
| A06:2021 Vulnerable & Outdated Component | High | 40 | 5 |
| | Medium | 40 | 5 |
| | Low | 0 | 0 |
| A08:2021 Software & Data Integrity Failures | High | 0 | 0 |
| | Medium | 0 | 0 |
| | Low | 53 | 20 |

### Funding
The authors received no funding for this work.

### Competing Interests
The authors declare that they have no competing interests.

## Author Contributions

- Sana Qadir conceived and designed the experiments, performed the experiments, analyzed the data, prepared figures and/or tables, authored or reviewed drafts of the article, and approved the final draft.
- Eman Waheed conceived and designed the experiments, performed the experiments, analyzed the data, performed the computation work, prepared figures and/or tables, and approved the final draft.
- Aisha Khanum conceived and designed the experiments, performed the experiments, analyzed the data, performed the computation work, prepared figures and/or tables, and approved the final draft.
- Seema Jehan conceived and designed the experiments, prepared figures and/or tables, authored or reviewed drafts of the article, and approved the final draft.

## Data Availability

The Yasca Reports are available at GitHub: https://github.com/daisy2310/Yasca-Reports.

The Progpilot Reports are available at GitHub:
https://github.com/daisy2310/Progpilot-Reports.

The Snyk Reports are available at GitHub:
https://github.com/daisy2310/Snyk-Reports.

The SonarQube Reports are available at GitHub:
https://github.com/daisy2310/SonarQube-Reports.

The OwaspZAP Reports are available at GitHub:
https://github.com/daisy2310/OwaspZAP-Reports.

The Wapiti Reports are available at GitHub:
https://github.com/daisy2310/Wapiti-Reports.

The Vega Reports are available at GitHub:
https://github.com/daisy2310/Vega-Reports.

The Iron WASP Reports are available at GitHub:
https://github.com/daisy2310/IronWASP-Reports.

The BurpSuite Reports are available at GitHub:
https://github.com/daisy2310/BurpSuite-Reports.

The list of targetWeb applications and their OWASP ZAP reports are available at GitHub: https://github.com/Aishakf7/OWASP-Based-Assessment-of-Web-Application-Security-Application-Reports-and-Application-links.

The reports are also all available at Zenodo: Qadir, S., Jehan, S., Khanum, A., & Emam, W. (2025). WebAppSecurity [Data set]. Zenodo. https://doi.org/10.5281/zenodo.15070993.

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
