# Peer review of "Comparative evaluation of approaches & tools for effective security testing of Web applications"

_PeerJ Computer Science, doi:10.7717/peerj-cs.2821_

## Round 0.1 · original submission · Major Revisions

Experts have now judged your manuscript and as you can read, their comments require major revisions. In particular, the reviewers require more explanation throughout and improved image quality, they report inconsistencies between some of the statements (e.g. in the Conclusion) and the research conducted, they ask to revisit RQ4, and they signal the lack of threats to validity and recommendations for future work. Please carefully revise your manuscript by taking into account all comments by the reviewers and make your study and presentation more rigorous by abiding to the well-known guidelines for performing empirical research.

Reviewer 1 ·

Basic reporting

The study mainly determines the effectiveness of SAST and DAST tools in finding OWASP Top
10:2021 and CWE Top 25:2023 vulnerabilities in web apps. For this purpose, 4 SAST and 5
DAST tools were selected to test 75 real-world web applications from technology, health and
education domains. The limitations of previous studies are highlighted in “Related Work” but it
is not mentioned that whether these limitations (some or all) are resolved in current study or not.

Experimental design

This work is specifically testing web applications but the RQs are just about the effectiveness of
DAST and SAST tools in finding OWASP Top 10 and CWE Top 25 vulnerabilities. Authors did
not mention about the web applications in any of the 4 RQs.
Need to add some details about “Report” column in Table 4. Which report it is representing?
Except 2, all SAST and DAST tools were deployed on Windows 10. Why these 2 tools were
deployed on Kali Linux. Any specific reason for using Linux for these tools?

Validity of the findings

In Results section, while answering “Research Question 1” authors have written “six out of nine
risk categories” while in “Research Question 2” it is “six out of ten”. Which one is correct?
In Table 8, the limitations of this work are not added.

Reviewer 2 ·

Basic reporting

The use of English is acceptable but still can be improved.
The introduction doesn’t clearly explain the urgency and research gap.
The figures are relevant but not well labeled and described. The quality is low.
The authors have provided the names and links of the tested websites, setup details, and the generated reports.

Experimental design

The study is within the scope of PeerJ Computer Science.
The research questions are well-defined, relevant, and meaningful.
There’s no ethical issue.
The method is not explained thoroughly and in detail.

Validity of the findings

The study has impact and novelty but is not replicable since the authors don’t explain how the approaches and tools are evaluated.
The authors could improve how they present data in the results section to make it easier to understand. The visualization should align with the purpose of the data visualization.
Several inconsistencies exist between the statements in the conclusion and the research conducted.

Additional comments

1. The abstract needs to be adjusted; it should contain the background, objective, method, result, discussion, and conclusion. The authors have not yet included the main discussion/findings that can answer the research objective.

2. Introduction:
The authors have explained the importance of security testing in the introduction section. However, the reason why evaluating approaches and tools for security testing is necessary has not been explained, so the urgency of this research is not clear.
The authors also need to explicitly state the research gap in this section and explain how this research addresses that gap. Authors can refer to related works to briefly explain the gap, such as Kunda and Alsamdi (2022), Li (2020), and Cruz et al. (2023).

It is recommended to cite sources for the statements in lines 44–45 and 49–50 to strengthen the arguments rather than presenting them as opinions.
The term "the former initiative" in line 50 is unclear. Specify whether it refers to CWE or OWASP, despite the citation that might clarify this.

3. Method
Why did the authors precisely formulate RQ4 to determine the consistency of findings in the OWASP ZAP tool?
The main concern with the method is that the authors have not explained how the evaluation process was conducted. Is this process carried out automatically by the tools?
The authors need to ensure that no other variables affect the evaluation results, as the vulnerability findings may also depend on the test design.
The authors must also define how ‘performance’/‘effectiveness’ is measured. Is it based on the number of vulnerabilities identified?
Clarify why you chose the other five tools, referencing previous research or explaining your rationale, as only four tools are discussed in lines 174–175.
In Table 3, add a row for the selected websites and specify that this study used 75 websites.
In line 180, explain how vulnerabilities and their severity are categorized. Are they tool-generated, validated by experts, or determined through another method?

4. Result:
The quality of the images needs to be improved.
The authors should consider the way the data is presented. It would be more appropriate if the authors used a regular bar chart instead of a stacked bar chart, as the goal is to compare the number of vulnerabilities found by each approach/tool, not to show proportions.
The use of labels is not consistent; on the Y-axis, the authors use percentages, but on the label of each bar, they use numbers. The authors need to add a label to clarify what the numbers on each bar represent.
Explain why you chose 50% as the threshold.
Reformat Table 7 to make it more readable. Consider adding a separate column for severity numbers instead of combining them with tool names, and use formatting to improve clarity.
Move lines 275–280 to the discussion section to explain the differences in findings for OWASP ZAP compared to Khanum et al. (2023).

5. Discussion:
In the discussion section, the authors must conduct an in-depth analysis based on the results to answer the research questions. In general, the authors only describe the results.
The authors can elaborate on which risk characteristics are suitable for each approach and tool, why a particular tool is more appropriate for risks with specific severity, and why this happens.
The author should acknowledge any limitations or threats to validity and provide specific, well-justified recommendations for future researchers.

6. In the conclusion (line 318), the authors state that there are 145 web apps, but in the method section, only 75 are mentioned.

---

## Round 0.2 · Major Revisions

Experts have now judged your revised manuscript and as you can read, there has been an improvement, but a few issues still remain to be resolved. In particular, (1) the Introduction needs to clearly present the research gap, (2) the evaluation process needs to be further clarified, particularly regarding how the methodology addresses the four research questions (RQs), and (3) the quality of the figures must be improved further, especially Figs. 3–5. Please carefully revise your manuscript by taking into account all remaining comments from the reviewers.

Reviewer 1 ·

Basic reporting

No comment

Experimental design

No comment

Validity of the findings

No comment

Additional comments

The authors were advised to mention the limitations of previous studies that are addressed in the
current study. They updated the "Related Work" section accordingly. Additionally, they were
asked to highlight the limitations of their own work. The authors clearly mentioned these limitations
in Table 6 as well as in other sections of the manuscript.
Furthermore, authors were suggested to revise the research questions and to provide a detail of
"Report" column in Table 2. They made appropriate improvements to the research questions and
added the requested detail of "Report" column.
A question was raised regarding the deployment of 2 tools on Linux and others on Windows 10.
They addressed this concern and included an explanation in the manuscript.
In short, the authors have resolved all the concerns and issues raised in first review.

Reviewer 2 ·

Basic reporting

The introduction does not clearly present the research gap.
The figures are relevant, but some are of low quality or appear blurry.
The authors have provided the names and links of the tested websites, setup details, and generated reports.
The discussion section could be improved.

Experimental design

The study is within the scope of PeerJ Computer Science.
The research questions are well-defined, relevant, and meaningful.
There are no ethical issues.
The methodology has been described in greater detail.However, the evaluation process remains unclear.

Validity of the findings

The study has both impact and novelty. The authors have stated that the dataset (raw data and results) can be accessed via a URL link. This information should be explicitly included in the data availability statement to ensure accessibility for readers.
Several inconsistencies between the statements in the conclusion and the research conducted have been resolved by the authors.

Additional comments

Abstract: The abstract has been revised to include a comprehensive summary of the background, objectives, methods, results, discussion, and conclusion.

Introduction: While the introduction section has seen significant improvements, the claim that the research gap is explicitly stated and addressed is still not convincing. The following sentence, for instance, requires further elaboration to highlight the research gap:
"The relevance and impact of these two lists is underscored by their use in multiple research studies, such as (Chaleshtari et al., 2023), (Shahid et al., 2022), (Li, 2020), and (Priyawati et al., 2022). Similarly, this research is a follow-up study of (Khanum et al., 2023) in which seventy Web applications were scanned using the OWASP ZAP tool to empirically investigate its effectiveness for detecting OWASP Top 10:2021 vulnerabilities."
This statement merely describes that the study will be conducted without specifying the improvements or contributions offered by this study.

Methods:

The methods section provides sufficient procedural details. However, additional context regarding the OWASP Top 10 and CWE Top 25 should be included to aid readers in understanding their relevance.
Questions remain about the evaluation process, particularly regarding how the methodology addresses the four research questions (RQs). The authors should clarify how results for each RQ are generated, categorized, and analyzed, including details about the evaluation process for each RQ.
Without proper evaluation methods, the validity of the study is difficult to confirm. For example, in the statement:
"Unfortunately, none of the tools were successful in finding vulnerabilities belonging to CWE-787 Out-of-bounds Write, CWE-416 Use After Free, CWE-125 Out-of-bounds Read, CWE-476 NULL Pointer Dereference, CWE-190 Integer Overflow or Wraparound, CWE-502 Deserialization of Untrusted Data, CWE-863 Incorrect Authorization, and CWE-276 Incorrect Default Permissions."
The authors claim that the tools failed to detect these vulnerabilities but do not clarify whether these vulnerabilities should have been present or were absent. Supporting evidence, such as annotated data or benchmarks, is necessary to validate this point.
Target Selection: It is unclear how the authors selected the target Web Apps. Did each Web App contain all the vulnerabilities? This needs to be explained to validate whether the tools were genuinely incapable (as suggested by row 301 in the results and the limitations in Table 6).

Figures and Tables:

The quality of the figures must be improved, especially for screenshots of web pages (e.g., Figures 3–5, which appear blurry).
Table formatting is incorrect; captions should be placed above the tables.
In Figure 8, only 9 out of 10 risk categories are displayed. Similarly, Figure 9 shows only 9 out of 25 risk categories. This discrepancy needs to be explained.
The meaning of "Both tools" in Figures 8 and 9 is unclear. Why do SAST/DAST tools alone detect vulnerabilities in some cases, while "both tools" fail, and vice versa? This inconsistency requires clarification.

Discussion: The discussion section could be more in-depth. It currently repeats the results without further analysis. A more effective structure would be to organize the discussion based on the findings for each RQ. This approach would clarify the study's contributions compared to existing research and ensure the analysis remains focused on addressing the RQs.

---

## Round 0.3 · accepted · Accept

As confirmed by the reviewers, the authors have made significant improvements in accordance with the concerns raised in the previous reviews. All major issues and concerns have now been addressed.

Reviewer 2 ·

Basic reporting

No comment

Experimental design

No comment

Validity of the findings

No comment

Additional comments

The authors have made significant improvements in accordance with the concerns raised in the previous review. All major issues and concerns have been addressed.